# The terminal Ediacaran Tongshan Lagerstätte from South China

Jin-bo Hou [1] ✉, Xiang-dong Wang[1], Zhang-shuai Hou[1], Jahandar Ramezani [2], Qing Tang[1] & Shu-zhong Shen [1] ✉

Because the informative Burgess Shale-type preservation is uncommon in the Ediacaran, mouldic Ediacara-type preservation provides insight into the early evolution of organisms like metazoans (including typical fronds), protists, and algae. Here, we report the Burgess Shale-type preservation from the new Tongshan Lagerstätte (~ 551–543.74 ± 0.87 Ma), in carbonaceous mudstones/shales of the terminal Ediacaran Dengying Formation in Tongshan, Hubei, South China. The preservation of a high diversity of organisms indicates rapid, likely in situ burial in the marine photic zone below the storm wave base, revealing deep-water biodiversity coeval with the Nama Assemblage of Ediacara Biota. These are the first records, to our knowledge, of typical Ediacaran rangeomorph fronds with Burgess Shale-type preservation. The presence of Burgess Shale-type preservation of fronds reflects the rarity of fine-grained deposits in the Ediacaran Period and bridges an important gap between traditional Ediacara-type preservation, Ediacara-type preservation with organic remains, and Burgess Shale-type preservation.

Much of our understanding of the Cambrian radiation, with the abrupt appearances of diverse metazoan genera, reflects the global distribution of fossil Lagerstätten in fine-grained deposits with Burgess Shale-type (BST) preservation[1–5]. This preservation type records compositional information in both chemical and morphological aspects of organic integuments such as cuticle details[6,7] and internal soft tissues such as muscles, gut systems, and nervous systems[8–11], in addition to hard exoskeletons. Other early metazoans occur in the Ediacara Biota of the late Ediacaran Period (~ 575–538.8 Ma), with the first appearance and diversification of macroscopic, multicellular and complex organisms[12–19]. Our understanding of these organisms is largely dependent on fossil Lagerstätten[18], and limited by the depositional environment and preservational mode of each deposit. Unlike Cambrian Lagerstätten with BST preservation, the Ediacara Biota features Ediacara-type preservation that preserves the outline of structures, but the compositional information in both chemical and morphological aspects is unavailable[20]. Extensive studies on the preservation of soft-bodied fossils have indicated that different styles of preservation, such as Ediacara-type, BST, and pyritization, bias our understanding of the early evolution of animals[21–25].

In this work, we document a new Lagerstätte from the terminal Ediacaran Dengying Formation in Tongshan County, Hubei, China. It exhibits a high diversity of organisms, including typical Ediacaran rangeomorph fronds, preserved within a short interval of carbonaceous mudstones/shales at the top of this formation. Its BST preservation type makes compositional information widely available to the Ediacaran metazoans, such as fronds.

## Results

### Locality of the Tongshan Lagerstätte

Abundant macrofossils were collected from the upper carbonaceous mudstones/shales of the siliceous rocks of the Dengying Formation at the Jiweijian and Wanjia sections in Tongshan County, Hubei Province, South China (Fig. 1a, b). The macrofossils are predominantly preserved with BST preservation. A stratigraphic horizon containing macrofossil *Shaanxilithes* occurs 18 meters above the

[1]State Key Laboratory of Critical Earth Material Recycling and Mineral Deposits, Frontiers Science Center for Critical Earth Material Cycling, School of Earth Sciences and Engineering, Nanjing University, Nanjing, China. [2]Department of Earth, Atmospheric and Planetary Sciences, Massachusetts Institute of Technology, Cambridge, MA, USA. ✉e-mail: hou@nju.edu.cn; szshen@nju.edu.cn

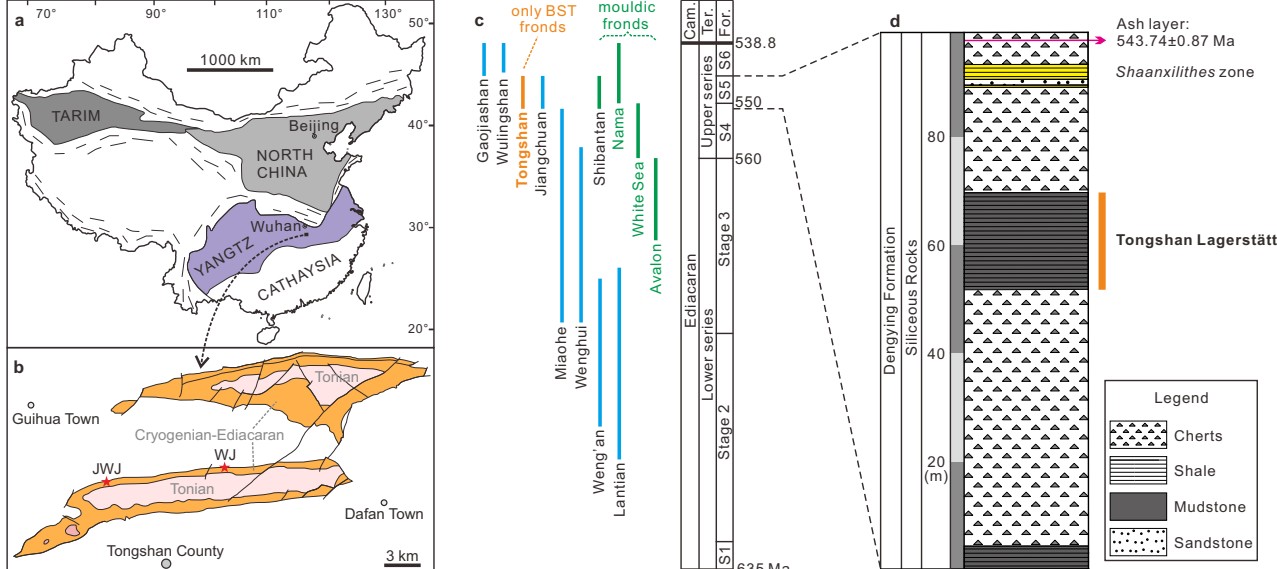

**Fig. 1 | Geological maps and stratigraphic scheme showing the Tongshan Lagerstätte. a** Simplified tectonic framework of China [Modified from[112] and permitted by Elsevier] showing the Yangtze Block that hosts the Tongshan Lagerstätte. **b** Distribution of the Neoproterozoic strata near Tongshan County in the Hefeng-Tongshan area, Hubei Province, South China; faults are displayed as thick lines; the Jiweijian (JWJ) and Wanjia (WJ) sections in Tongshan County are marked with red stars. **c** Correlation of the Ediacaran biotas in China and the three assemblages (Avalon, White Sea and Nama) of the Ediacara Biota. **d** Detailed stratigraphy of the upper part of the siliceous rocks of the Dengying Formation at Wanjia section, Tongshan County. Tongshan Biota is marked with orange bar and the age of zircons is dated from the ash layer collected at the Jiweijian section, Tongshan County. Cam., = Cambrian; For., Fortunian; Ter., Terreneuvian.

Tongshan Lagerstätte at the Wanjia section and is also present at the Jiweijan, Gaojiashan and Zhelinziwan sections (Supplementary Figs. 1a–e, 2). A volcanic ash layer ~4.8 m above the *Shaanxilithes* zone at the Jiweijian section (Supplementary Fig. 1e) has produced a weighted mean $^{206}$Pb/$^{238}$U zircon date of $543.74 \pm 0.87$ Ma (Fig. 1d, Supplementary Figs. 1e and 3) by the high-precision chemical abrasion isotope dilution thermal ionization mass spectrometry (CA-ID-TIMS) of this study. The latter date establishes, by correlation, a minimum age constraint for the Tongshan Lagerstätte, as well as for the Shibantan and Jiangchuan Lagerstätten (Fig. 1c, Supplementary Fig. 1). An ash layer from the uppermost Miaohe Member of the underlying Doushantuo Formation in the Yangtze Gorge area (Jiuqunao section) was dated at $551.1 \pm 0.7$ Ma by the CA-ID-TIMS method[26,27], which, by correlation to the Wuhe section, places a maximum age constraint on the Shibantan Lagerstätte. Similarly, the Tongshan Lagerstätte can be bracketed in age between ca. 551 Ma and $543.74 \pm 0.87$ Ma. The Tongshan Lagerstätte becomes age-equivalent to the early phase of the Nama Assemblage of the Ediacara Biota[28], an interval marking the earliest diversification of biomineralized metazoans[29].

## Organisms of the Tongshan Lagerstätte

Macroalgae are the main component of the Tongshan Lagerstätte with seven known taxa: *Baculiphyca taeniata* (Fig. 2a, Supplementary Fig. 4a–c), *Chuaria* sp. (Fig. 2g), *Doushantuophyton lineare* (Fig. 2c),? *Gesinella hunanensis* (Supplementary Fig. 4d), *Longifuniculum dissolutum* (Fig. 2j), possible *Tawuia* (Supplementary Fig. 4e), and *Zhongbaodaophyton robustus* (Fig. 2b). Five undescribed forms have been discovered: a *Doushantuophyton*-like form (Fig. 2d), a baseball bat-like form with an expanded distal end (Fig. 2e), an irregular circular form (Fig. 2f) with bounded filaments, a net-like form (Fig. 2i) with crossing filaments, and a ribbon-like form (Fig. 2k) with bounded filaments. The undescribed filamentous macroalgae (Fig. 2d, f, i, k) are larger than previously known taxa, other than the erect, skinny *Doushantuophyton*. *Beltanelliformis brunsae,* with a possible microbial origin (Fig. 2h, Supplementary Fig. 5a–d), and the enigmatic

*Palaeopascichnus linearis* (Fig. 3c, Supplementary Fig. 6a), are also present. Four putative metazoans include three known species, *Protoconites minor* (Fig. 3a), *Sinospongia chenjunyuani* (Supplementary Fig. 6b–c), *Eoandromeda octobrachiata* (Fig. 3d), and one undefined tubular form with segment-like divisions (Fig. 3b). This youngest report of *Eoandromeda* extends its occurrence from the White Sea Assemblage to the Nama Assemblage, suggesting the previously known disappearance of octoradialomorphs near 551 Ma[30] is a taphonomic artifact or sampling bias. A tubular structure (Fig. 3b) showing segment-like divisions and its distal portion showing an irregular opening at the top may be related to either cnidarians or annelids by comparison to tubular cloudinomorphs.

Typical frondose organisms discovered in the Tongshan Lagerstätte, including Rangeomorpha clade[31], have been phylogenetically linked with metazoans[13,17,19,32,33]. The specimen ESEN 0016, with a distally tapering end that curves to the upper right side, is clearly divided longitudinally into three distinct lobe-like regions: a slightly wider central region and two lateral regions (Fig. 4a, c). The two lateral regions represent two individual petaloids, each composed of dozens of the first-order branches arranged parallel to one another (Fig. 4b, d). The second-order branches, visible under low-angle lighting, are almost perpendicular or slightly inclined relative to the first-order branches (Fig. 4b). The central region contains the stalk; however, this region likely includes a few overlapping petaloids, which are difficult to distinguish due to compression. The overlapping stalk and petaloids are more distinct at the proximal part of the specimen, where two lateral petaloids are arranged in parallel along the long axis of the central stalk. The total body construction and the presence of the first- and second-order branches indicate this specimen is a typical rangeomorph metazoan.

Ten fan- and bush-like specimens are similar to *Bradgatia* (Fig. 5a, b). Segments of branches are clearly visible and show 3D in morphology (Fig. 5d–f). The fan-like shape with V form (Fig. 5a) is comparable to the early growth stage of *Bradgatia*[34]. The possible elliptical holdfast at the proximal end of the bush-like form (Fig. 5b, c) provides evidence that the attachment style is comparable to the holdfasts of other

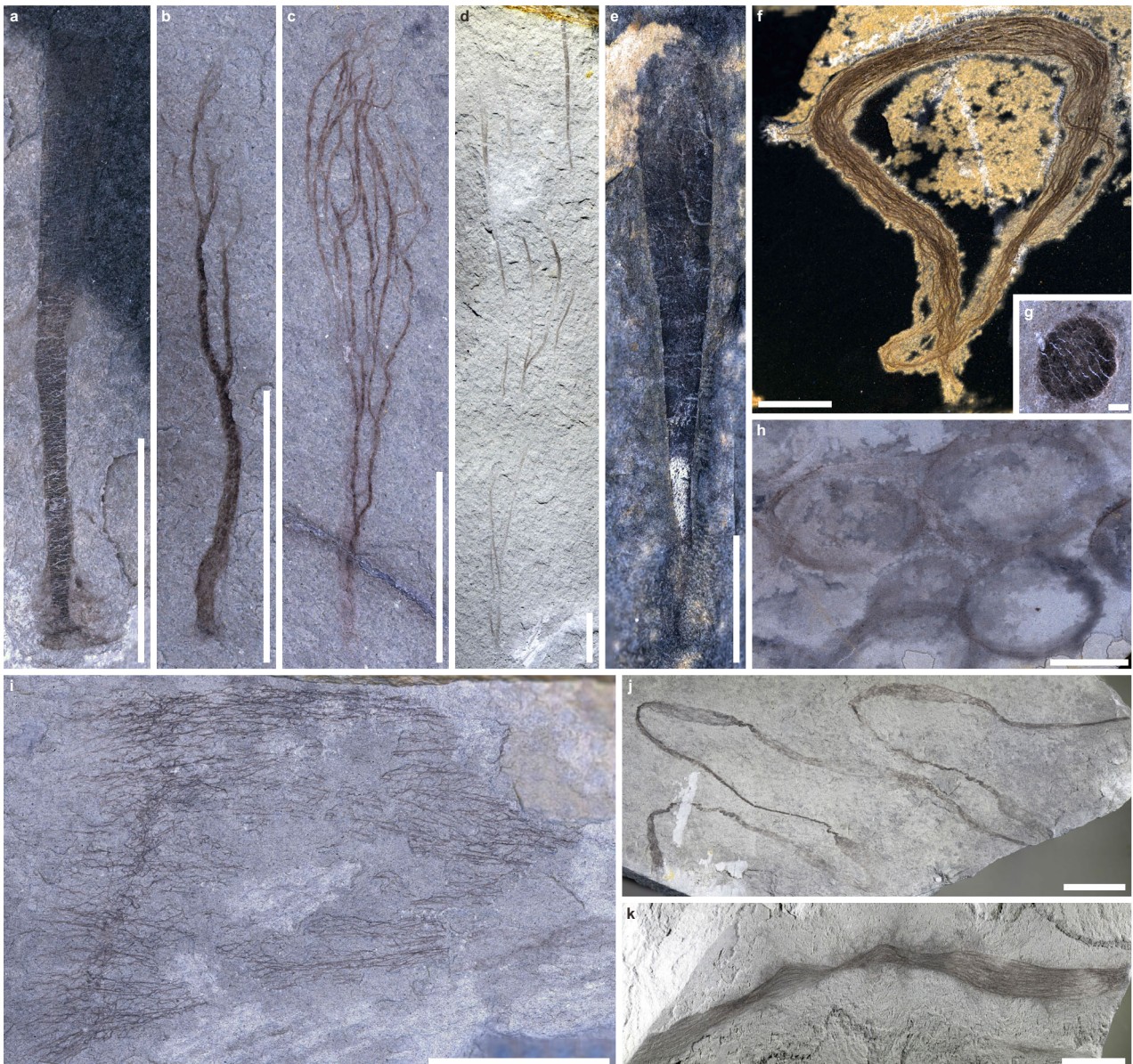

**Fig. 2 | Carbonaceous macroalgae and microbes of the Tongshan Lagerstätte.** **a** *Baculiphyca taeniata*, ESEN 0001. **b** *Zhongbaodaophyton robustus*, ESEN 0002. **c** *Doushantuophyton lineare*, ESEN 0003. **d** Undefined *Doushantuophyton*-like form, ESEN 0004. **e** Undefined baseball bat-shaped form, ESEN 0005. **f** Undefined irregular circular form, ESEN 0006. **g** *Chuaria sp.*, ESEN 0007. **h** *Beltanelliformis brunsae*, ESEN 0008. **i** Undefined net-like form, ESEN 0009. **j** *Longifuniculum dissolutum*, ESEN 0010. **k** Undefined ribbon-like form, ESEN 0011. Scale bars in **a**–**f** = 5 mm, in **g** = 0.5 mm and in **h**–**k** = 10 mm.

typical Ediacara fronds. Ontogenetically *Bradgatia* changes from forms I to V to U and to O[24], the two forms (Fig. 4a, b) become comparable to the intermediate growth stages (V and U) of *Bradgatia*. All evidence indicates they are rangeomorphs.

Dozens of specimens, such as ESEN 0019 (Fig. 5g–h) have a spherocylindrical outline with a blunt distal end are comparable to rangeomorphs such as *Charniodiscus* based on broad morphology. The elliptical holdfast may reflect lateral compression during preservation. The width of the petalodium is equal to the diameter of the holdfast. Contrary to one petaloid on the left side, the two almost parallel longitudinal curves on the right side are likely an impression of the margins of two individual petaloids. The first-order branches are parallel to one another (Fig. 5i), while the second-order branches are perpendicular to or inclined relative to the first-order branches (Fig. 5j–k). Some annulation-like divisions on the second-order branches may represent the potential third-

order branches (Fig. 5j–k). A circular spot, defined by a faint white margin at the center of the holdfast, represents the stem or stalk of the petalodium (Fig. 5h). As the petalodium and holdfast were compressed during preservation, whether a stem is present remains unclear. However, if a stem was originally present, the stem must have been short.

Another form with five specimens has the widest portion located on the lower part of its fusiform body (ESEN 0020; Fig. 6a–d). This form has a long tail-like distal end that is similar to *Paracharnia*. The first-order branches are parallel to each other (Fig. 6a–d) and some second-order branches (Fig. 6c) are distinct near the proximal portion of some primary branches. At the middle portion, the first-order branches of the petaloid meet at the mid-line (Fig. 6a–b). On the right side, a few parallel curves may represent the margins of several superimposed individual petaloids (Fig. 6b). Petaloid branches are clearly visible on both sides even if the whole body is slightly twisted.

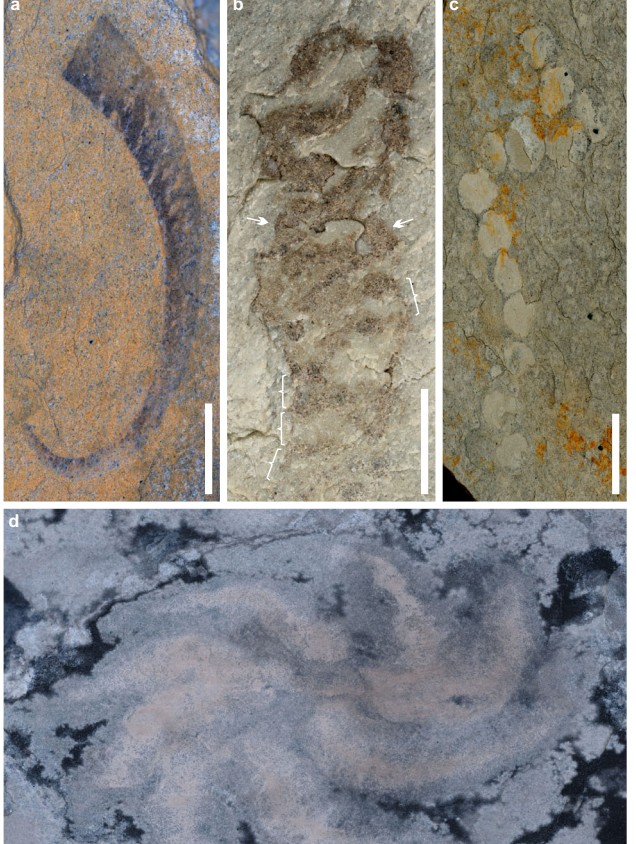

**Fig. 3 | Putative metazoans and foraminifers of the Tongshan Lagerstätte.**
**a** Carbonaceous *Protoconites minor* with a possible cnidarian affinity, ESEN 0012.
**b** BST preservation of a tubular fossil with segment-like divisions (marked with white brackets) has a possible metazoan affinity, ESEN 0013; white arrows pointed to an elliptical cross-section of a segment. **c** BST *Palaeopascichnus linearis* with a possible foraminifer affinity, ESEN 0014. **d** BST *Eoandromeda octobrachiata*, with a possible ctenophore affinity, showing eight arms and a centrally located aboral pole, ESEN 0015. Scale bars in **a**, **c** and **d** = 5 mm and in **b** = 2 mm.

The proximal, elliptical end represents the holdfast because it stands out from the stalk region at the upper left corner. All these characters indicate it is a typical rangeomorph.

One specimen has a blunt conical outline, with one petaloid on the left side and possibly two superimposed petaloids on the right side (ESEN 0021; Fig. 6e–g). Parallel segments of petaloids are clearly recognizable on the upper part of the specimen. A possible holdfast with a faint outline is preserved with a diameter about half of the maximal petalodium diameter (Fig. 6g). A circular spot located at the center of the possible holdfast represents the stem or stalk of the petalodium (Fig. 6g). These characters indicate it is a typical frondose organism.

EDS analyses of two frond specimens, ESEN 0017 and ESEN 0019a, display high concentrations of C and Fe in the body fossils, which contrast with the high concentrations of Mg, Al, K, Ti, and Ba in the matrix (Table 1, Supplementary Fig. 7). In both specimens, Si and O exhibit similar concentrations in both the fossil body and the matrix. Specimen ESEN 0017 (Supplementary Fig. 7a) has a higher concentration of S in the fossil body, whereas specimen ESEN 0019a (Supplementary Fig. 7l) has a higher concentration of P (Table 1, Supplementary Fig. 7).

## Discussion

### Setting of the Lagerstätte

Sedimentary evidence reveals the depositional setting of the Tongshan Lagerstätte. Thin sections of shales/mudstones show parallel laminae composed of thick clay-rich background beds and thin silt-rich event beds (Supplementary Fig. 8). The thick clay-rich beds indicate a prolonged period of low-energy conditions, alternating with short-duration silt-rich beds formed under slightly higher energy. Periodic fluctuations in local environmental energy resulted in the rapid burial of organisms. In addition, these parallel beds lacking sedimentary features such as cross-stratification indicate that the deposition occurred below the storm wave base. Together, the carbonaceous mudstones and shales hosting the Tongshan Lagerstätte represent a quiet, relatively deep setting below storm wave base, likely related to marine transgression[35]. This deep setting complements the shallow-water setting of the Nama Assemblage (Fig. 1c), extending biodiversity of the latest Ediacaran organisms[36] (Supplementary Fig. 6d) to the deep marine environments (Supplementary Information). The late Ediacaran marine transgression of the Yangtze Block was associated with widespread upwelling of anoxic water from the deep ocean to the shallow marine settings[37]. Paleogeographic reconstruction[35] positions the Tongshan Lagerstätte on the marginal slope of the siliceous basin or in the transitional zone between the carbonate platform and the siliceous basin[38–41] (Supplementary Fig. 9a, b). Tectonic subsidence associated with an extensional fault across the middle Yangtze Block (Supplementary Fig. 9a)[35] resulted in a steep, south-dipping, transitional zone, where the Tongshan biota was located, with deposits different from the carbonate platform to the north. Diverse macroalgae (Fig. 2) indicate that this setting was in the photic and oxic zone, similar to the settings in which the Miaohe and Lantian Lagerstätten were situated[12,42]. Here, the putative sedentary metazoans (Figs. 3a–d, 4a–d, 5a–j, 6a–g, Supplementary Fig. 6a–c) evidently had sufficient oxygen and nutrients to flourish (Supplementary Information).

### Preservation of rangeomorph fronds

The metazoans from the Dengying Formation described here are either attached or lying on the bedding surface. Here, we present small individual fronds (Figs. 5, 6), with a body length less than 30 mm, that have been rarely documented elsewhere[43–46]. As branches were added distally during development, small individuals record the earliest growth history of the fronds[44], thus providing an opportunity to explore their growth mode[43]. Greater size or height is correlated with increased dispersal capacity, suggesting the potential for colonization may have been emphasized more than resource competition[47]. Thus, small individuals from the Tongshan Lagerstätte likely represent rapid colonization of fronds on carbonaceous muds.

In Lagerstätten with characteristic Ediacara-type preservation, holdfasts are rarely preserved on the same horizon as their fronds; in most localities, either one or the other is preserved[24]. But most material described here has both holdfasts and their associated fronds preserved on the same horizon. With respect to the Ediacara-type preservation that has a size limitation due to the relatively coarse grain size of the casting medium[45,46], small particles of mudstones capture exquisite structural detail, as is typical in BST fossils[9]. In the Ediacara-type specimens, structures are hard to distinguish from the surrounding matrix as boundaries between body fossil and matrix are blurred, but BST specimens have a distinct contrast from the surrounding matrix and well-preserved structural details. This is one of the reasons why the holdfasts can be observed in our materials. The high concentration of Fe in the body fossil (Table 1, Supplementary Fig. 7a) indicates the exceptional preservation of soft-bodied fossils because pyrite, formed by iron reacting with sulfates, can quickly replace the soft tissues at the early stage of decay[4,21,48,49], allowing the preservation of soft tissues[50]. The low proportion of pyrite in fonds makes the Tongshan Biota most comparable to the preservation

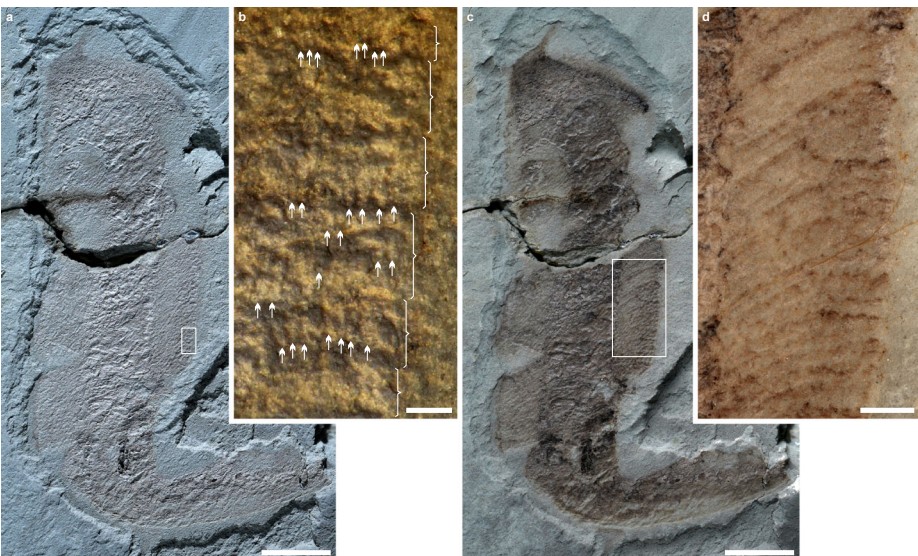

**Fig. 4 | Burgess Shale-type preservation of a typical Ediacara frond in the Tongshan Lagerstätte. a**–**d** Undescribed rangeomorph, ESEN 0016, about 52 mm in length. **b** Close-up of the area taken with low-angle lighting in the white box of figure a showing the first-order branches marked by brackets and the second-order branches pointed out by white arrows. **c** Specimen spread with alcohol. **d** Close-up of the area within the white box in figure c showing the first-order branches that are parallel to one another. Scale bars in **a** and **c** = 5 mm, in **b** = 0.2 mm and in **d** = 1 mm.

condition of the middle Cambrian Burgess Shale Lagerstätte (Supplementary Information). The high concentration of P in the specimen ESEN 0019a (Table 1, Supplementary Fig. 7l) provides additional chemical evidence for the early diagenetic mineralization of soft tissues[51]. Phosphatization preserves tissues with high fidelity but is a strongly biased taphonomic process affected by factors such as size and environment[52,53]. Decay of the small-sized frond may produce a suitable microenvironment for the process of phosphatization. The complete and exquisite preservation, such as the BST fronds with both petalodium and holdfast, the macroalgae with both thallus and holdfast, aggregated *Beltanelliformis* without overlapping, and the geochemical data, indicates quick burial, likely in situ preservation.

**Expansion of the Ediacara Biota with BST preservation**

Contrary to the conventional view[54], several lines of evidence suggest that reliance upon Ediacara-type preservation for analyzing the Ediacaran Period has biased our understanding. Previous studies have not, to our knowledge, documented BST preservation of fronds. Partial carbonaceous preservation of the only charnid specimen from the Ust-Pinega Formation of Siberia[55] suggests the holdfast was made of resistant material[24] and was preserved as carbonaceous compressions with their soft stems replicated by pyrite[24]. A single specimen of *Charnia masoni* on the surface of an enigmatic carbonaceous compression from the Khatyspyt Formation of Siberia has been interpreted as a superimposed taphonomic phantom of the fossil, indicating that the conditions favoring carbonaceous preservation were selective against rangeomorph and frondomorph tissues[56]. This denotes that the exclusion of the Ediacara-type *Charnia* and *Hiemalora* from the Miaohe-type (also the BST) preservational window is a real taphonomic signal that provides an important constraint on the properties of Ediacaran organismal tissues[56], and that the BST window appears to select against many Ediacara-type taxa[25]. Some BST fronds preserved in the Cambrian, which lack typical characters such as first- and second-order branches, are cautiously described as Ediacaran-like fronds[57] or problematica[58]. The preservation of these debated Cambrian organisms[58,59] fails to provide solid evidence for the preservation of those typical Ediacara fronds. Taken together, typical Ediacara fronds have not been recovered with BST preservation in either the Ediacara Biota or other Ediacaran BST biotas (Supplementary Fig. 10a).

BST preservation of typical fronds indicates that Ediacara-type preservation, which largely occurs in coarse-grained deposits such as sandstones, is restricted by available rocks. A low delivery rate of continental clays to Ediacaran oceans[60,61] may have resulted in the paucity of BST Lagerstätten during the Ediacaran (Supplementary Information), lowering sampling intensity and affecting the biodiversity of Ediacaran organisms[30] in fine-grained deposits such as mudstones. As the iron weight percentage of sediment (iron: sediment) plays an important role in Ediacara-type preservation and is generally higher in shales than in sandstones[50], preservation bias for Ediacaran organisms is further highlighted by the rarity of available deposits, such as mudstones from the Ediacaran period. Even in such a globally harsh geological background for exquisite fossil preservation, the Tongshan Lagerstätte still indicates that exploring more BST Lagerstätten in the Neoproterozoic is now a practical and promising task.

BST preservation of typical fronds opens a window onto the expansion of the Ediacara Biota and the exploration of the earliest metazoans and stem-group organisms (Supplementary Fig. 10b). Recent documentations of soft-bodied Ediacara-type holdfasts in the early Cambrian (Fortunian and Stage 2)[62,63] extend Ediacara-type preservation/organisms across the Ediacaran-Cambrian boundary. The material described here extends the documentation of BST-style preservation into other elements of the Ediacara Biota, demonstrating a continuation from the Cambrian organisms to the Lantian Biota[14,64] as well as between the Ediacaran-like fronds in the Cambrian and typical Ediacaran fronds. Laboratory simulations have indicated that fossils of the Ediacara Biota preserve not only the external morphology but also the morphology of soft external or internal organic skeletons[65]. Also, presence of organic matters has played an important role in the preservation of fine details of early metazoans through the Ediacara-type preservation[65]. Analytical techniques such as biomarkers and stable isotopic data have helped to solve long-standing debates by revealing compositional information of tissues[66–70]. The BST preservation of the Ediacara Biota reinforces these observations and offers greater opportunities to examine internal skeletons or structures in detail, contributing to solving long-standing debates related to the early evolution of metazoans.

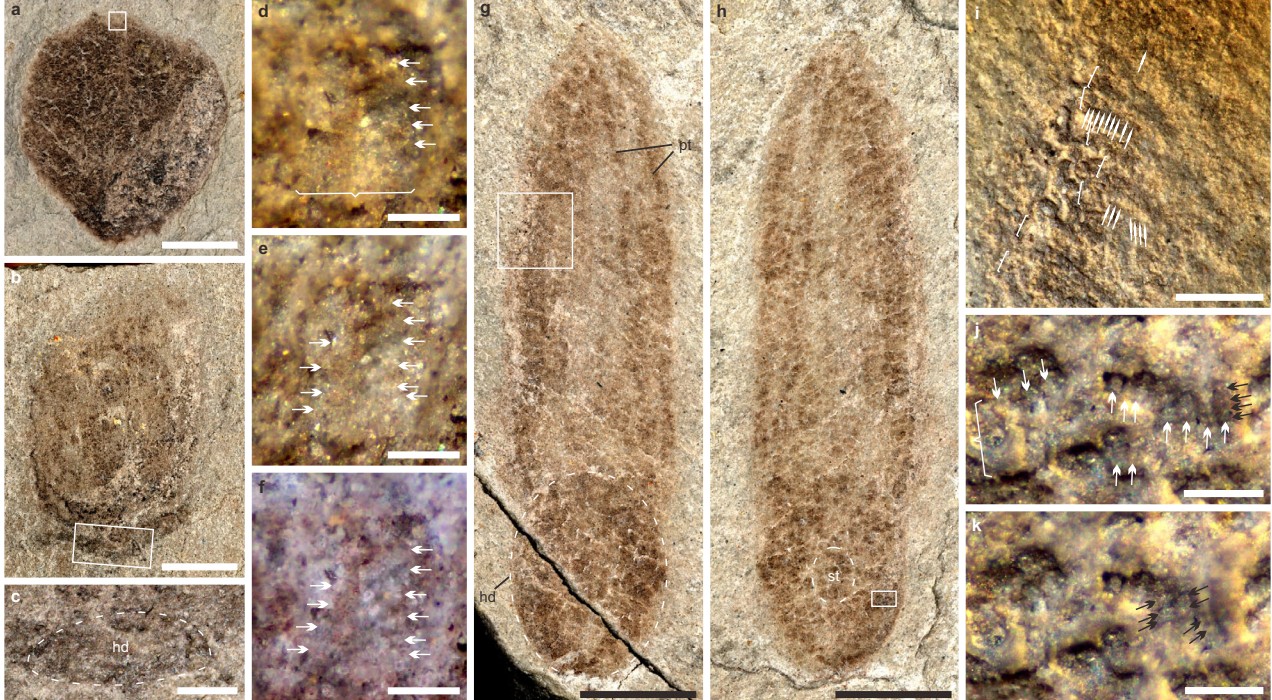

**Fig. 5 | Burgess Shale-type preservation of the typical Ediacara fronds in the Tongshan Lagerstätte. a** Fan-shaped form comparable to *Bradgatia*, ESEN 0017, about 6.2 mm in length. **b** U-shaped form comparable to *Bradgaita* ESEN 0018, about 7.5 mm in length. **c** Close-up of the area within the white box of figure b showing the possible holdfast outlined with an elliptical dashed line. **d–f** Close-up of the area within the white box in figure a showing the first-order and second-order branches. **d** Image with low-angle lighting showing the second-order branches on the right side of the first-order branch. **e** Image with a different lighting direction showing the second-order branches on the left side of the first-order branch. **f** a stacked image with polarized light showing the second-order branches. **g–h** Undescribed rangeomorph, part ESEN 0019a and counterpart ESEN 0019b, about 11.5 mm in length. **g** Holdfast outlined with an elliptical dashed line. **h** Stem outlined with an elliptical dashed line. **i** close-up of the area within the white box of figure g showing the parallel first-order branches and second-order branches. **j–k** Close-up of the area within the white box of figure h showing the first-order, second-order, and possible third-order branches. **j** A stacked image showing the first-order, second-order, and possible third-order branches. **k** Image with low-angle lighting showing the possible third-order branches. The first-order branches are marked with white brackets. The second-order branches are pointed out by white arrows. The possible third-order branches are pointed out by black arrows. hd, holdfast; pt, petaloid; st, stem or stalk. Scale bars in **a–b** and **g–h** = 2 mm, in **c** and **i** = 0.5 mm and in **d–f** and **j–k** = 0.1 mm.

## Methods

### Fieldwork
We acquired permission from the landowner for conducting scientific research. The late Ediacaran Dengying Formation is consistently and widely distributed in the Tongshan County and adjacent areas. The first fossil was discovered at the Jiweijian section, which is also where tuff sample was dated. After extensive exploration, we selected the Wanjia and Jiweijian sections, Tongshan County, Hubei, China, for detailed study. Fossils were collected and carefully packed in the field and transported to the School of Earth Sciences and Engineering at Nanjing University between summer 2022 and spring 2025.

### Dengying Formation in Tongshan County, Hubei
Ediacaran strata in Hubei Province of South China are stratigraphically and palaeogeographically divided into the Yichang-Tongshan and Yunxian-Yingshan regions, of which the Yichang-Tongshan region is further subdivided into large Huangling-Dahongshan and small Hefeng-Tongshan areas[71]. Extensive studies have been carried out in the Huangling-Dahongshan area[72–74], but little work has focused on the Hefeng-Tongshan area. The Dengying Formation in the Hefeng-Tongshan area is divided into two parts; the lower limestones are 74.08 m thick, whereas the upper siliceous rocks are 224.75 m thick[71] (Supplementary Fig. 1c). This is in contrast to the adjacent Huangling-Dahongshan area, including the Yichang and Three Gorge Area, where the Dengying Formation is subdivided into the Hamajing, Shibantan, and Baimatuo Members in ascending order (Supplementary Fig. 1b)[73].

Lower limestones of the Dengying Formation in Tongshan County are dominated by limestones and dolostones (Supplementary Fig. 1c). Upper siliceous rocks of the Dengying Formation have the basal part dominated by limestones and dolostones and the upper part composed of four lithological units: dark carbonaceous mudstones, thin- to medium-thick gray bedded cherts, dark gray carbonaceous mudstones, and thin- to medium-thick bedded cherts, in ascending order[71] (Fig. 1d, Supplementary Fig. 1c). Our materials were yielded from and through upper carbonaceous mudstones/shales at the Jiweijian and Wanjia sections of the Tongshan County (Fig. 1a, b, d, Supplementary Fig. 1c–e), about 350 km away from the Three George Area where the Miaohe and Shibantan Biotas were discovered[42,73]. The fossil-bearing strata are continuous in the Hefeng-Tongshan area (Fig. 1b, Supplementary Fig. 1c–e). As the materials were first discovered at the Jiweijian section, Tongshan County, Hubei Province, South China, we here name this fossiliferous deposit the Tongshan Lagerstätte.

*Shaanxilithes* (Supplementary Fig. 1f), a potential index fossil defining the base of the Ediacaran Stage 6[75–78], has a wide geographical occurrence in Hubei, Guizhou, Ningxia, Qinghai, Shaanxi, and Yunnan provinces of China[79,80], northwestern India[77], central and southeastern Siberia[81–83], and possibly southern Namibia[84] (Supplementary Fig. 2). The consistent appearance of *Shaanxilithes* in Tongshan County strengthens its potential of an index fossil by its massive occurrence over a short stratigraphic interval about 4 m in thickness (Fig. 1d, Supplementary Figs. 1d, e, 2). The Tongshan Lagerstätte is located at about 18 m below the 4 m thick siliceous shales containing the genus

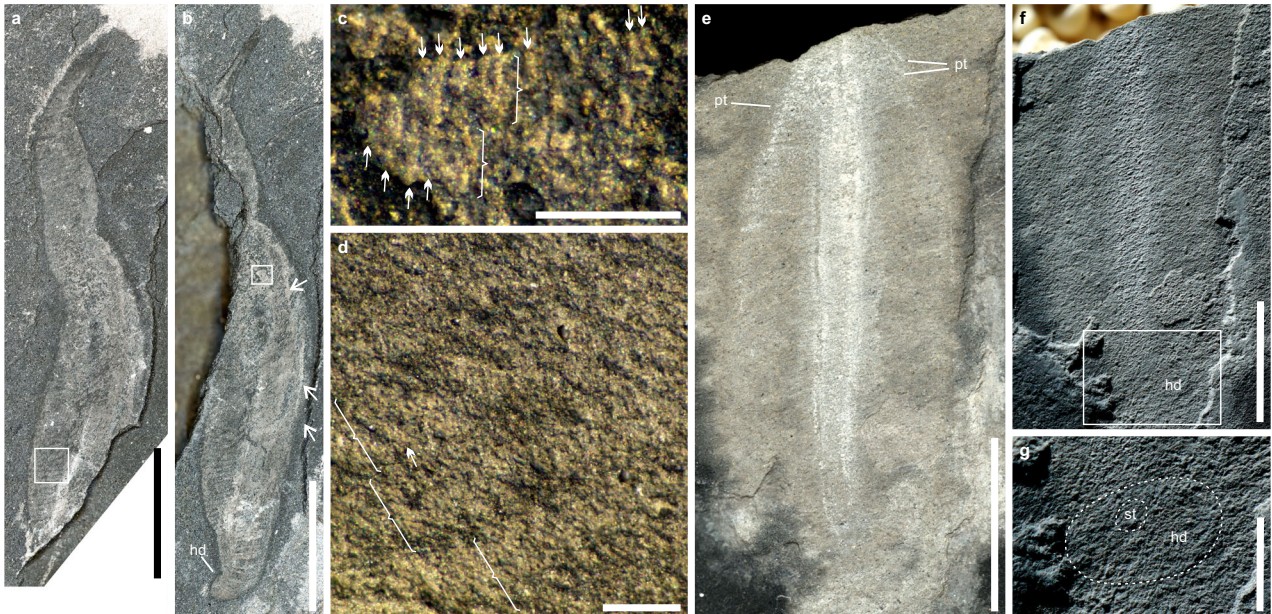

**Fig. 6 | Burgess Shale-type preservation of the typical Ediacara fronds in the Tongshan Lagerstätte. a** Unnamed rangeomorph, counterpart ESEN 0020b (left-right reversed image). **b** Unnamed rangeomorph, part ESEN 0020a, about 22.8 mm in length; arrows pointed to the superimposed individual petaloids. **c** Close-up of the area within the white box of figure b showing the first-order (marked with brackets) and second-order (pointed out by white arrows) branches. **d** Close-up of the area within the white box of figure a showing the first-order (marked with brackets) branches. **e** Unnamed frond, ESEN 0021, about 16.1 mm in length. **f** Specimen photographed with a low-angle lighting to show a faint holdfast. **g** Close-up of the area within the white box of figure f showing the possible holdfast (the external dashed line) and a central stem (the inner dashed line). hd, holdfast; pt, petaloid; st, stem or stalk. Scale bars in **a–b** and **e–f** = 5 mm, in **d–d** and **g** = 0.2 mm.

*Shaanxilithes* (Supplementary Fig. 1c–e). In the nearby Yangtze Gorges area of western Hubei Province, *Shaanxilithes* occurs in the uppermost 3.1 m interval of the Shibantan Member[80], of which the lower half hosts the Shibantan Lagerstätte (Supplementary Fig. 1b)[73,80]. The upper carbonaceous mudstones/shales of the Denying Formation in the Hefeng-Tongshan area of southern Hubei are thus correlated to the Shibantan Member in the Yangtze Gorges area of western Hubei, and to the lowermost part of the Gaojiashan Member in the Ningqing area, Shaanxi Province (Supplementary Fig. 1), assuming a synchronous *Shaanxilithes* zone. Hence, the Tongshan Lagerstätte is most likely time equivalent to the Shibantan Lagerstätte (Fig. 1c, Supplementary Fig. 1). As the *Shaanxilithes* zone is stratigraphically directly above the Jiangchuan, Shibantan, and Tongshan Lagerstätten, our high-precision U-Pb tuff age of 543.74 ± 0.87 Ma (Supplementary Fig. 2) from the Jiweijian section of Tongshan County may serve as a minimum age constraint for the three Lagerstätten. Since the Gaojiashan Lagerstätte in Shaanxi Province of China occurs above the *Shaanxilithes* biozone, the Tongshan and Shibantan Lagerstätten are likely older than the Gaojiashan Lagerstätte (Fig. 1c, Supplementary Fig. 1). The relatively few fossil Lagerstätten in the Neoproterozoic results in poor stratigraphic correlations and difficulty in understanding the true fossil ranges. Temporal ranges on the order of 10 Ma or longer for the Ediacaran biota or assemblages (Fig. 1c)[85], compared to just about 2–3 Ma for the Cambrian ones[86], limit our ability in understanding the evolutionary patterns and processes prior to the Cambrian explosion. Further high-precision geochronology from a variety of the Ediacaran Lagerstätten is required to address the problems regarding the evolution of multicellular life in the Neoproterozoic.

## Social network analysis
Comparing related and adjacent biotas provides important information for understanding how palaeocommunities changed in space and time. Macro-organisms documented in the Lantian, Miaohe, Wenghui, Shibantan, Jiangchuan, Wulingshan, and Tongshan Lagerstätten are

compiled in Supplementary Data 2. Social network analysis was performed in RStudio version 1.3.959.

## Elemental mapping
Two frond specimens were selected for scanning electron microscopy-energy dispersive X-ray spectroscopy (SEM-EDS) analysis using ZEISS GeminiSEM 360 attached with ULTIM MAX detector. For each specimen, two sites on the frond body and one site on the matrix were selected (Supplementary Fig. 7a, l). All sites were defined with a uniform area of approximately 1.12 mm×0.84 mm, the data of which are presented in Table 1. An additional site, the fourth site, outlined with a red box in each specimen (Supplementary Fig. 7a, l), displays the difference in elemental distribution between the frond body and the matrix.

## Zircon U-Pb geochronology
The tuff sample (JWJ23-D-01) from the Jiweijian (JWJ) Section, about 4.8 m above the *Shaanxilithes* zone, was analyzed by the U-Pb CA-ID-TIMS method, following the detailed procedures described by Ramezani et al. [87]. Zircon separates were extracted from the ash sample using standard crushing, magnetic susceptibility, and high-density liquid techniques. Most zircons are prismatic with delicate glass (melt) inclusions parallel to their long axis under binocular microscope, suggesting a volcanic origin[87].

The selected zircon grains were pretreated by a chemical abrasion procedure modified after Mattinson[88], which involved thermal annealing in a 900 °C furnace for 60 h, followed by partial dissolution (leaching) in 29 M HF inside high-pressure Parr® vessels at 210 °C for 12 h in order to mitigate Pb-loss effects, which often lead to anomalously young dates. The partially dissolved grains underwent fluxing alternately with dilute $HNO_3$ and 6 M HCl on a hot plate and in an ultrasonic bath, each for 1 h. After each ultrasonic step, the grains were rinsed with ultra-pure water to remove the leachates. Subsequently, the thoroughly rinsed zircon grains were spiked with the EARTHTIME ET535 mixed $^{205}Pb$-$^{233}U$-$^{235}U$ tracer[89,90] before complete dissolution in

**Table 1 | Summary of EDS spectra of two frond specimens**

| Element | ESEN 0019a | | | ESEN 0017 | | |
|---|---|---|---|---|---|---|
| | Site 1 | Site 2 | Site 3 | Site 1 | Site2 | Site 3 |
| C | 9.4 | 8.7 | 8.5 | 8.5 | 9.5 | 5.6 |
| Fe | 2.2 | 1.9 | 1.1 | 1.0 | 0.9 | 0.8 |
| O | 52.1 | 51.9 | 51.6 | 47.4 | 47.9 | 47.4 |
| Si | 20.1 | 20.7 | 21.4 | 27.7 | 26.7 | 25.8 |
| Mg | 0.7 | 0.8 | 0.8 | 0.7 | 0.7 | 0.9 |
| Al | 9.0 | 9.6 | 10.1 | 8.9 | 9.0 | 11.5 |
| K | 3.8 | 4.1 | 4.5 | 4.1 | 3.9 | 5.6 |
| Ti | 0.3 | 0.4 | 0.7 | 0.3 | 0.3 | 1.1 |
| Ba | 0.8 | 0.8 | 0.9 | 1.1 | 1.1 | 1.4 |
| P | 1.5 | 1.1 | 0.1 | | | |
| S | | 0.1 | 0.2 | 0.2 | 0.1 | |
| Cr | | | | 0.1 | | |

Numbers represent relative concentration (weight percentage or wt%) of elements. All sites (including site 1 and site 2 on the body, and site 3 on the matrix) were retrieved from a uniform area. One experiment was performed for each site.

29 M HF at 210 °C for 48 h. The zircon solutions were then dried down on a hot plate and redissolved in 6 M HCl inside high-pressure vessels at 180 °C overnight. The dissolved U and Pb were chemically purified by an HCl-based column chemistry method using AG1X-8 anion-exchange resin. The eluted U and Pb were dried down with 0.05 M $H_3PO_4$ and loaded with a silica gel emitter solution onto a zone-refined outgassed Re filament for mass spectrometry.

The U and Pb isotopic ratios were measured on an Isotopx® X62 multi-collector thermal ionization mass spectrometer equipped with a Daly photomultiplier ion-counting system at the Massachusetts Institute of Technology Isotope Laboratory. Pb isotopic ratios were measured as monoatomic Pb ions in a peak-hopping mode on the ion-counter, whereas U isotopes were measured as $UO^{2+}$ in a static mode on three Faraday detectors simultaneously. Measured isotopic ratios were corrected for mass-dependent isotope fractionation in the mass spectrometer, as well as for U and Pb contributions from the spike and laboratory blanks. Common Pb in the analyses averaged 0.29 pg, all of which was attributed to laboratory blank, and its isotopic composition was determined from long-term measurements of the total procedural Pb blank in the lab (see Supplementary data 1 footnotes). The radiogenic $^{206}Pb$ concentrations were also corrected for initial $^{230}Th$ disequilibrium in zircon using a magma Th/U model ratio of $2.8 \pm 1.0$ ($2\sigma$).

Data reduction, calculation of dates and propagation of uncertainties were carried out using the Tripoli version 3.7.1 and ET_Redux version 0.5.2[91,92]. Uncertainties in the calculated weighted mean $^{206}Pb/^{238}U$ date are reported at $2\sigma$ level (Supplementary Fig. 3) and in the $\pm X/Y/Z$ Ma format, where $X$ is the internal (analytical) uncertainty in the absence of all external errors, $Y$ incorporates $X$ and the U–Pb tracer calibration errors, and $Z$ includes the latter as well as the decay constant errors of[93]. The external uncertainties must be taken into account only if the results are compared with U–Pb dates obtained in other laboratories with different tracers, different techniques (e.g., microbeam U–Pb), or ones derived from other isotopic chronometers (e.g., $^{40}Ar/^{39}Ar$).

All five analyzed zircons from the tuff sample JWJ23-D-01 overlap within their $2\sigma$ analytical uncertainties, yielding a weighted mean $^{206}Pb/^{238}U$ date of $543.74 \pm 0.87/0.98/1.1$ Ma with a MSWD of 1.0. This date provides the best age estimate for the eruption of the tuff and a good approximation for the associated sedimentary depositional age.

## Ages of Ediacaran Lagerstätten

The Ediacaran-Cambrian (E-C) boundary provides the lower age limit for comparison of the Ediacaran Lagerstätten. High-resolution CA-ID-

TIMS geochronology from Namibia brackets the E-C transition between $539.18 + 0.17/ - 0.26$ Ma and $538.30 + 0.14/ - 0.14$ Ma[94] or between $538.99 \pm 0.21$ Ma and $538.58 \pm 0.19$ Ma[95], although the index fossil *Treptichnus pedum* was not recorded in any of the calibrated biostratigraphies. Defining the E-C boundary is outside the focus of this study; we here consider 538.8 Ma for the E-C boundary following the geologic time scale[16,94].

Wulingshan Lagerstätte (545–538.8 Ma). Wulingshan Lagerstätte was erected in 1999 based on collections from the middle portion of the Liuchapo Formation at Ligonggang town, Taoyuan County, western Hunan, South China[96]. The Liuchapo Formation at Ligonggang town has a total thickness of 39.8 m and is subdivided into the lower 6.8 m in thickness composed of siliceous rocks and carbonaceous siliceous slates, the middle 23 m in thickness with carbonaceous shales, and the upper 10 m in thickness composed of argillaceous siltstones and thin layered siliceous rocks. Without index fossils such as *Shaanxilithes* in the Liuchapo Formation, it is hard to make a stratigraphic correlation between the Tongshan and Wulignshan Lagerstätten. While the fossils described by Chen et al.[96] are macroalgae[97], the absence of metazoans makes it difficult to correlate with the Tongshan Lagerstätte. By considering both Lagerstätten that share a high similarity of macroalgal forms, we suggest that they are stratigraphically close to each other. An ash layer among the cherts of the lower Liuchapo Formation in west Hunan has yielded a U-Pb CA-ID-TIMS age of $545.76 \pm 0.66$ Ma[98], suggesting that the Wulingshan Lagersätte is likely younger than 545 Ma[99]. Accordingly, we here consider its age between ~545 Ma and 538.8 Ma (Fig. 1c).

Shibantan Lagerstätte (previously known as Xilingxia Lagerstätte or Shibantan Assemblage; 551–544 Ma). The name complexity of the Xilingxia Fauna is discussed by Xiao et al.[26] and we here follow Xiao et al. in using the Shibantan Lagerstätte. The minimum age of the Shibantan Lagerstätte[26] has been considered $543.4 \pm 3.5$ Ma based on the U-Pb zircon age of a tuffaceous layer from the Baimatuo Member in the Yangtze George area above the *Shaanxilithes* zone[100]. This age assignment is consistent within uncertainty with our minimum age of $543.74 \pm 0.87$ Ma for the Tongshan Lagerstätte mentioned above (Fig. 1c).

Jiangchuan Lagerstätte (551–544 Ma). Jiangchuan Lagerstätte was discovered from the Jiucheng Member of the Dengying Formation in Jiangchuan and Jinning, eastern Yunnan, South China[101,102]. *Shaanxilithes* occurs in the Xiaowaitoushan Member of the Zhujiaqing Formation in eastern Yunnan, stratigraphically far above the Jiucheng Lagerstätte[102], which correlates the Jiangchuan Lagerstätte possibly to the Tongshan and Shibantan Lagerstätten.

Miaohe Lagerstätte (> 551 Ma). Miaobe Lagerstätte was formally erected in 1992 by Ding and his colleagues[103]. The Miaohe Member, from which the Miaobe Biota was discovered, is either correlated to the Doushantuo Member IV[67,104] or directly below the Shibantan Member of the Dengying Formation[105]. The recent data from the Miaohe Member suggest that the Miaohe Member is most likely correlated to the Doushantuo Member IV because of a slipping block incorporating both the Miaohe Member of the Doushantuo Formation and the Hamajing Member of the Dengying Formation[106] (Fig. 1c).

Wenghui Lagerstätte (586–557 Ma). The Wenghui Biota[107–109] at the Xiajiaomeng section in the Guizhou Province of South China occurs below an ash layer with a U-Pb zircon age of $557 \pm 3$ Ma[110]. We here follow Yang et al.[110] to assign an age range to the Wenghui Biota slightly shorter than that of the Miaohe Biota. This biota has a maximum age of $585.7 \pm 2.8$ Ma[110] (Fig. 1c).

## Image collection and preparation

The specimens were photographed using Olympus DSX1000 with DSX10-SXLOB10X lens, Nikon SMZ25 with SHR Plan Apo 0.5x lens, and Sony ILCE-7RM4A with 50 mm lens. Olympus DSX1000 was equipped with a polarized filter. Both low-angle lighting under

normal optical conditions and polarized light were used to capture different structural details of the specimens. Alcohol spreading on specimens was also applied in some conditions to enhance the contact between fossil and rock matrix. The continental reconstructions generated using GPlates 2.3.0 were modified from the plate model (Version 1.1b with a file name of SM2-4485738-V2) in Merdith et al. [111]. Figure 1b is based on a geological map sourced from the publicly accessible GeoCloud platform and was created using Adobe Illustrator 26.0.

## Reporting summary

Further information on research design is available in the Nature Portfolio Reporting Summary linked to this article.

## Data availability

Fossil specimens (ESEN 0001 – ESEN 0031) and tuff sample (JWJ23-D-01) described in this paper are deposited in the School of Earth Sciences and Engineering at Nanjing University (ESEN) in China and are available for further research by contacting the corresponding authors. All study data are included in the article, supplementary information, and supplementary data.

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

## Acknowledgements

This work was supported by the International Postdoctoral Exchange Fellowship Program of the China Postdoctoral Science Foundation (YJ20220270 to J.-B.H.), the GeoX Interdisciplinary Project of Frontiers Science Center for Critical Earth Material Cycling (20250101 to Q.T., and 20250201 to J.-B.H.), and the Fundamental Research Funds for the Central Universities (020614380234 and 020614380237 to Q.T.). We thank Xu-dong Hou, Yu-kun Shi, Hai-peng Xu, Bao-yu Jiang, and Li Zhang of Nanjing University for technical help; Tan-xi Hou's family for their support of fieldwork; Douglas H. Erwin, Nigel C. Hughes, and Mary L. Droser for comments, and edits, that have much improved this manuscript.

## Author contributions

J.-B.H. and S.-Z.S. designed and supervised this work. J.-B.H. collected and photographed specimens, and prepared figures. S.-Z.S., Z.-S.H.,

and J.R. conducted the zircon dating. J.-B.H. and S.-Z.S. wrote the initial manuscript, with input from X.-D.W., Z.-S.H., J.R., and Q.T.

## Competing interests

The authors declare no competing interests.
