## [Transparent Peer Review file · Nature Communications]

The terminal Ediacaran Tongshan Lagerstätte from South China

Corresponding Author: Dr Jin-bo Hou

Version 0:

Reviewer comments:

Reviewer #1

(Remarks to the Author)

This paper documents a new late Ediacaran-aged lagerstätte from China, that preserves a mixture of algae, macroscopic protists, and apparent animals as carbonaceous impressions. The style of preservation offers a unique view into the morphology of several taxa, and – because it occupies a deeper-water setting than many contemporaneous deposits – the lagerstätte as a whole provides an interesting window into latest Ediacaran ecosystems that isn't preserved in many other places.

This is a nicely written paper that I think should be published. It currently suffers from a couple of weaknesses that muddy the waters somewhat...but I'm confident that the authors can fix these without too much extra work. I've outlined these major points below, and then subsequently some more minor comments that should be easy to remedy.

1. The algal and protistan fossils presented in the paper are beautifully preserved, and will provide the basis for fascinating research for years to come. However, I am less convinced by many of the metazoan (and supposedly frondose) fossils...and I think the authors could usefully take a step back from over-interpreting these. For example: with respect to the fossils shown in Figure 4a-d, even after combing through the images in the supplementary material I cannot see any convincing rangeomorph elements, and thus I think the identifications are highly speculative. I am similarly unconfident about the evidence for holdfasts...and without more examples I think it's very hard to say anything about mode of life, or even what these fossils represent. Similarly, the fossil shown in Figure 4e shows little resemblance to *Swartpuntia*; for a start *Swartpuntia* possesses a segmented (or at least, 'ornamented') stem, and the spacing of putative erniettomorph tubular modules would be highly unusual for this taxon. This is a lengthy way of saying: I think it would be much more sensible to refer to these as 'putative frondose metazoans' – the paper doesn't rely on these to be *Bradgatia* and *Swartpuntia* to be publishable in *Nature Communications*, and referring these fossils to specific Ediacaran genera distracts from what is otherwise a very exciting find. In place of this discussion, I think instead the paper would be strengthened with more information given on the preservation (see below):

2. 'Burgess shale-type' preservation has been defined a few different ways, but is perhaps most commonly used to refer to soft-tissues preserved as carbonaceous impressions, sometimes with associated pyrite, and often with clay minerals that may or may not be diagenetic in origin. It would be fascinating if the authors could show some data pertaining to the preservation and lithological context of these new fossils – for example, it would be great if some SEM images or EDS elemental map data could be given, and which would allow the authors to compare their style of preservation to BST and other instances of carbonaceous compression fossils. Lastly, I think some pictures of the outcrop would be valuable; this would allow the reader to eyeball the sedimentology, and would help support the paleoenvironmental interpretation that is being offered.

Minor comments:

"Unlike many Cambrian Lagerstätten, Burgess Shale-type (BST) preservation is rare and moldic Ediacara-type preservation provides insight into the early evolution of metazoans including typical fronds" (line 15); this sentence appears to be saying two different things – I think it needs re-wording.

"The large surface area of the new algae might have locally increased oxygen concentration via photosynthesis to facilitate metazoans" (line 87); this reads a little strangely, and feels like an unsupported statement. I suggest adding a reference, or

removing if there is no previous work to support this being important in Ediacaran ecosystems.

"...whether a stem is present remains unclear but present, the stem must have been short" (line 144); likewise, this sentence reads a little strangely...do the authors mean to write, 'if a stem was originally present, it must have been short'?

Reviewer #2

(Remarks to the Author)

At present, the submitted paper describes Burgess Shale type preservation as that 'which preserves original information on organic integument and internal soft tissues, in addition to hard exoskeletons.' The paper then states 'Unlike Cambrian Lagerstätten with BST preservation, the Ediacara Biota 43 features Ediacara-type preservation that preserves the outline of structures, but the original compositional information is unavailable.'

It is correct that many Ediacaran biota fossils are preserved as moulds or casts. However, while rarer than mouldic preservation, organic preservation of Ediacaran fossils is known for example from the White Sea assemblage. E.g. see Bobrovskiy et al 2019 Nature Ecology and Evolution:

<https://www.nature.com/articles/s41559-019-0820-7>

'Preservation of organic matter plays an important role in the taphonomy of all fossils from the White Sea localities. A close investigation of dozens of Ediacaran macrofossils, including Dickinsonia, Andiva, Kimberella, Aspidella, Sabelliditides and Beltanelliformis collected from unweathered layers in the White Sea revealed that they all are preserved organically, including those that occur within sandstone or clay (Fig. 6a–d)6,7,45.'

See also Cai et al 2012 Palaeo³:

<https://www.sciencedirect.com/science/article/pii/S003101821200082X>

'Burgess Shale-type (BST) fossils often are preserved as two-dimensional carbonaceous compressions, sometimes aided by two mineralization processes: pyritization and aluminosilicification, defined by a thin and sometimes localized coating of authigenic pyrite or aluminosilicate minerals on the carbonaceous materials. Here we report similar mineralization modes within the late Ediacaran Gaojiashan Lagerstätte of southern Shaanxi Province, South China.'

Therefore the manuscript should be updated in the light of known Ediacaran organic preservation.

Unfortunately, the majority of the figure photographs do not show sufficient detail of the specimens to confirm their taxonomic attributions.

For example the manuscript states 'Ten fan- and bush-like specimens belong to Bradgatia 104 sp. (Fig. 4a–b and Extended Data Fig. 7a–f) are one of the youngest appearances of this genus^{12,28 105}. Segments of branches are clearly visible (Extended Data Fig. 106 7a–f) and show 3D in morphology (Extended Data Fig. 7e–f).'

However, figure 4 a and b show circular specimens with some irregular lines. If we compare these with Ediacaran specimens with exceptional preservation, that show their taxonomically distinctive features it is very difficult to confirm whether they may be of the same taxon or not.

Similarly in extended data fig 7 there is a large amount of textural noise in the images and it is not possible to confirm the presence of, for example, rangeomorph type fractal branching.

The fossil shown in Fig 4 c to d is very interesting. However, from these images it is difficult to confirm more than they show an elongate bag-like organism with at least first order branches, ribs or annulations.

Fig 4e is compared to Swartpuntia. However, this taxon is notable for having a wide aspect ratio which the figured specimen does not appear to share. What is the basis for the suggested attribution to this taxon as opposed to another taxon entirely? The discussed holdfast is not clearly visible in the figures.

Overall, I would suggest that there are two options available to the authors. First, either to improve, if possible, the images used to support the taxonomic attributions and to further explain and evaluate the evidence for specific attributions. Second, to make more tentative taxonomic attributions in the present paper. Perhaps a combination of these options could be pursued.

Reviewer #3

(Remarks to the Author)

In the Abstract (line 28) the authors state that their fossils will provide 'A new repository of original compositional information'. Likewise in the Introduction (line 49) 'This study ... makes compositional information widely available to the Ediacaran metazoans such as fronds' and Discussion (final line 238) 'the preservation ... will reveal more information on the original composition of Ediacaran metazoans'. However, this claim, which the authors appear to consider central to the significance of their paper, is nowhere supported by any chemical data. Note also that the composition will not be 'original' but diagenetically altered to longer chain compounds by polymerization.

'Bat-shaped', 'tie-shaped' (lines 83,84), and 'corn-like' (line 136) are not useful descriptors as they will suggest different comparisons to different people. Likewise 'meandering-like' (line 94) is not going to prompt images of this specimen.

The surface area (line 88) would only be a significant source of oxygen if the algae occurred in dense groups. Is there any evidence of this?

Does 'undefined' (line 94) simply mean that the specimen in 3b is poorly preserved. Likewise (line 99) how confident are you that this specimen is an open tube or that the irregular opening represents the true termination.

The designation 'sp.' means that the material preserves features diagnostic of the genus yet cannot be assigned to a known species. If you use 'sp.' you need to justify its assignment to the genus, but also explain how it differs from other species of the genus in question. You achieve this to some extent in the case of *Swartpuntia* (line 126 ...) but not for *Bradgatia* (line 103 ...).

Fresh algae and animals can endure significant transport without damage. The preservation of holdfasts does not necessarily demonstrate that the organisms are buried in situ. Is there sedimentological evidence to support 'quick burial and in situ preservation' (line 179)?

You need citation(s) to the conventional view that there is no bias (line 202).

If you cite 53 on line 237 (Love et al. 2009) you should also cite the alternative more recent view: Bobrovskiy, I., Hope, J.M., Nettersheim, B.J., Volkman, J.K., Hallmann, C. and Brocks, J.J., 2021. Algal origin of sponge sterane biomarkers negates the oldest evidence for animals in the rock record. *Nature Ecology & Evolution*, 5(2), pp.165-168.

Version 1:

Reviewer comments:

Reviewer #1

(Remarks to the Author)

The paper is substantially improved - the authors have backed off the specific genus designations, and have presented some new specimens which are more convincingly rangeomorphs. I am also grateful that elemental maps are shown (in Supp. Fig. 7), which give valuable context. If I was being nitpicky, the figure doesn't seem to present maps of fossils vs. matrix; some important information is thus hidden, and preventing a proper comparison. Ideally, one of the maps would encompass the border between the fossil and matrix, so enrichments in specific elements are clearer.

Otherwise, I think this is close to being publishable. Two final comments are below:

"pyrite, formed by iron reacting with sulfates, can quickly replace the soft tissues at the early stage of decay, allowing the preservation of soft tissues" (line 183); the authors may wish to cite Gibson et al. (2023) in *Geobiology*, who were able to experimentally replicate the formation of pyrite in association with decaying diptoblasts and triptoblasts.

"BST preservation of typical fronds indicates that Ediacara-type preservation, which largely occurs in coarse grained deposits such as sandstones, is restricted by available rocks" (line 239); similar to the comment above; Gibson et al. (2023) showed that the iron content of sediments is likely also an important control on Ediacaran-style preservation, and which hints at secular and paleogeographic biases.

Reviewer #2

(Remarks to the Author)

As detailed in their point by point response to review the authors have provided additional information and revised the text of their manuscript describing a fascinating new fossil biota. However, I would still suggest they go further in re-framing their discoveries to be less dependent on specific affinities of some of the fossils to Ediacaran taxa, which remain uncertain. The authors have provided a new figure 4, which they link to Rangeomorpha. However, given the difficulty in assessing branching structure from photographs, I am unable to verify the presence of rangeomorph elements.

I would suggest rephrasing the text throughout to suggest plausible affinities, and not to frame the paper in terms of novelty dependent on the more uncertain of these affinities, which I don't believe is necessary for this to be a very interesting and publishable paper.

For example, I would suggest modifying line 17 of the abstract to briefly describe the whole diversity of fossils they have found and the range of their plausible affinities e.g. all the protists, algae, metazoans they have identified, and mentioning apparent fronds within this without focusing on them so heavily.

Instead of 'representing the Rangeomorpha' in the abstract, I would suggest briefly including some of the more nuanced discussion of the BST preservation which the authors included in their point by point response e.g. summarising their response:

'Our Tongshan material exhibits true BST preservation (carbonaceous 244 compression in fine-grained mudstones/shales).'
'The BST Tongshan 248 Lagersätte bridges the gaps between traditional Ediacara-type preservation, Ediacara249 type preservation with organic remains (as seen in Bobrovskiy et al. 2019), and BST 250 preservation.'

I would suggest similar re-framing throughout.

Reviewer #3

(Remarks to the Author)

The authors have dealt with two major issues raised by the reviewers. The quality of the illustrations is greatly improved and a new specimen added. The evidence for fractals is difficult to demonstrate but is now much clearer. The taxonomic assignments are more generalized to reflect the nature of the material.

The authors have added EDS data (Figure S7 and Table 1). Table 1 provides weight per cent data but there is no information in Methods indicating whether these are based on the area mapped or spot samples. Unfortunately Figure S7 includes maps of one site on each specimen only whereas three were mapped. It would be useful to include the maps for the other areas (particularly the matrix). Needless to say maps convey different information to weight percent data, specifically element distribution, and overlap of different elements provides an indication of the minerals present.

Given that the EDS data are important to the interpretation of the preservation (BST-type or not) that information should be moved from Discussion to Results. Note that the high concentration of Fe in the body fossil does not, per se, indicate exceptional preservation, and the evidence for pyrite is weak at best: Fe and S coincide somewhat in Figure S7 C and K, but not in N and V (indeed S is essentially absent in V). Likewise the fact that the lithology is dominated by silicates does not allow a silica (i.e. quartz) cement to be inferred: the silicates are presumably clays as petrography of the thin sections would confirm. A high concentration of P is mentioned (line 180) but there is no discussion of its significance and a map of P is omitted for the first frond specimen in Figure S7!

Given the thickness of the silt laminae (Figure S8) it is hard to imagine rapid burial of in situ fronds.

The message in Figure S10 is not clear (to this reviewer at least). BST preservation is already included in the Ediacaran in MacGabhann's original (2014) diagram, which is identical to A here, so what does this figure add.? And coloring most the Ediacaran in green in B obscures information rather than adding to it.

Finally, even though the authors have removed mention of 'original' chemical composition Reviewer 3's concerns about the inclusion of phrases such as 'informative repository of tissue and integument composition' and 'compositional information in both chemical and morphological aspects' remain. This issue needs to be clarified so that it is obvious that what is meant is the composition of the fossils, not the composition of the original tissues. This confusion is exemplified in the authors' response where they state that 'In contrast, in BST preservation, the spine or claw can show multiple internal structural layers (e.g., *Hallucigenia sparsa* in Smith and Ortega-Hernandez, 2014, *Nature*, 514, 363-366), which can further be distinguished based on their chemical compositions, appearing as dark- and light-colored regions.' That paper includes no chemical data and it is very unlikely that the dark and light colored regions differ significantly in either their original (presumably chitin-protein) or diagenetically altered state.

Version 2:

Reviewer comments:

Reviewer #2

(Remarks to the Author)

My only further suggestion is to consider whether there are similarities to Cambrian fronds and to provide a reference to previous work that has identified fronds in BST preservation modes. The classic reference, for example is Conway Morris, S (1993) Ediacaran-like fossils in Cambrian Burgess Shale-type faunas of North America. *Palaeontology* 36, 593–635.

This was also recently reviewed in

Hoyal Cuthill, J.F., 2022. Ediacaran survivors in the Cambrian: suspicions, denials and a smoking gun. *Geological Magazine*, 159(7), pp.1210-1219.

Reviewer #3

(Remarks to the Author)

The authors have addressed the concerns of this reviewer.

Manuscript ID: **NCOMMS-25-0235-T**

We sincerely thank the reviewers for their thoughtful and constructive comments, which
have been instrumental in strengthening the scientific rigor of our manuscript. In response
to their valuable feedback, we have conducted a thorough revision of the text, significantly
expanded our dataset, and refined our interpretations. We have toned down the results
and discussions about the BST frond animals.

Key improvements include:

1) restricting frond materials to rangeomorphs;

2) adding additional analytical data, including a new table (Table 1) and two
supplementary figures (Figs. S7, S8), providing critical geochemical and sedimentological
evidence that elucidates the preservation of this Lagerstätte;

3) enhancing visual documentation, with all frond specimens—including a newly
added specimen (Fig. 4)—rephotographed and presented as individual figures (Figs. 4–
6). Low-angle lighting techniques were employed to better resolve fine morphological
details, such as first- and second-order branches;

4) providing the new Supplementary Figure S10, which shows the expansion of BST
(Burgess Shale-type) preservation in Ediacaran fronds and the Ediacara Biota, reinforcing
the broader implications of our findings.

All modifications are clearly highlighted in the revised manuscript (yellow background),
and point-by-point responses to each comment are provided below.

24 **Editorial feedback**

* Please replace any bar graphs with plots that feature information about the distribution
of the underlying data. All data points should be shown for plots with a sample size less
than 10. For larger sample sizes, please consider box-and-whisker or violin plots as
alternatives. Measures of centrality, dispersion and/or error bars should be plotted and
described in the figure legend.

Response:

Thank you for recognizing this detail. The vertical red bars in the Supplementary Figure
S3 represent the rank order plot, which is the standard format for reporting the isotope
dates (Condon et al. 2024, Recommendations for the reporting and interpretation of
isotope dilution U-Pb geochronological information, GSA Bulletin, 136, 4233–4251). It
should be noted that each bar represents one individual zircon, which is comparable to
the “individual points are shown when possible” in the Editorial Policy Checklist.

We have corrected the caption to become “The conventional concordia plot for
sample JWJ23-D-01 shows individual zircon analyses with their 2σ error ellipses, and all
individual analyses intersect the concordia line and are therefore concordant. The inset
rank-order plot of the zircon $^{206}\text{Pb}/^{238}\text{U}$ dates shows individual zircon age results with 2σ

uncertainty, and the horizontal black line with shaded bands signifies the calculated
weighted mean $^{206}\text{Pb}/^{238}\text{U}$ age with 2σ error (X/Y/Z). z1 to z5 in the figure represent five
zircon crystals analyzed using the CA-ID-TIMS method.”.

**REVIEWER COMMENTS**

Reviewer #1 (Remarks to the Author):

This paper documents a new late Ediacaran-aged lagerstätte from China, that preserves
a mixture of algae, macroscopic protists, and apparent animals as carbonaceous
impressions. The style of preservation offers a unique view into the morphology of several
taxa, and – because it occupies a deeper-water setting than many contemporaneous
deposits – the lagerstätte as a whole provides an interesting window into latest Ediacaran
ecosystems that isn't preserved in many other places.

This is a nicely written paper that I think should be published. It currently suffers from a
couple of weaknesses that muddy the waters somewhat...but I'm confident that the
authors can fix these without too much extra work. I've outlined these major points below,
and then subsequently some more minor comments that should be easy to remedy.

**Response:**

We sincerely appreciate your thoughtful assessment of our work and your recognition of
the significance of this newly discovered late Ediacaran Lagerstätte. We have carefully
addressed the major concerns raised in this point, which we acknowledge as critical to
strengthening the manuscript. Below, we provide a detailed point-by-point response to
your comments.

1. The algal and protistan fossils presented in the paper are beautifully preserved, and will
provide the basis for fascinating research for years to come. However, I am less convinced
by many of the metazoan (and supposedly frondose) fossils...and I think the authors could
usefully take a step back from over-interpreting these. For example: with respect to the
fossils shown in Figure 4a-d, even after combing through the images in the supplementary
material I cannot see any convincing rangeomorph elements, and thus I think the
identifications are highly speculative. I am similarly unconfident about the evidence for
holdfasts...and without more examples I think it's very hard to say anything about mode
of life, or even what these fossils represent. Similarly, the fossil shown in Figure 4e shows
little resemblance to Swartpuntia; for a start Swartpuntia possesses a segmented (or at
least, 'ornamented') stem, and the spacing of putative erniettomorph tubular modules
would be highly unusual for this taxon. This is a lengthy way of saying: I think it would be
much more sensible to refer to these as 'putative frondose metazoans' – the paper doesn't
rely on these to be Bradgatia and Swartpuntia to be publishable in Nature
Communications, and referring these fossils to specific Ediacaran genera distracts from
what is otherwise a very exciting find. In place of this discussion, I think instead the paper
would be strengthened with more information given on the preservation (see below):

Response:

We have included a new specimen (now Fig. 4) that exhibits clearer fractal branching,
reinforcing the presence of rangeomorph-type architecture in the Lagerstätte. This larger
specimen help clarify the branching structures in Figs. 5–6. The fractal patterns in several
forms of fossils are consistent with rangeomorphs. In accordance with your suggestion,
we have removed definitive assignments to Bradgatia and Swartpuntia and now refer to
these fossils as "rangeomorphs" or "frondose metazoan" throughout the text, emphasizing
their uncertain affinity. Some taxonomic names of frondose animals mentioned in the main
text are solely for comparative purposes.

2. 'Burgess shale-type' preservation has been defined a few different ways, but is perhaps
most commonly used to refer to soft-tissues preserved as carbonaceous impressions,
sometimes with associated pyrite, and often with clay minerals that may or may not be
diagenetic in origin. It would be fascinating if the authors could show some data pertaining
to the preservation and lithological context of these new fossils – for example, it would be
great if some SEM images or EDS elemental map data could be given, and which would
allow the authors to compare their style of preservation to BST and other instances of
carbonaceous compression fossils. Lastly, I think some pictures of the outcrop would be
valuable; this would allow the reader to eyeball the sedimentology, and would help support
the paleoenvironmental interpretation that is being offered.

Response:

We have added two supplementary figures (S7 and S8) and one table (Table 1) to support
the discussion of preservation. Figure S7 displays the elemental maps of two frond
specimens, and a summary of EDS spectra of both specimens is provided in Table 1.
Figure S8 contains one field image and two thin sections.

As you have mentioned, we found high concentration of Fe in the body fossils,
which is a good indicator of the exquisite preservation. The well-preserved laminated
beds of mudstone/shales provide strong evidence of an environment that experienced a
long duration of low-energy conditions, alternated with short duration of slightly high-
energy conditions.

In the main text, we add "*EDS analyses of two frond specimens display high*
*concentrations of C, Fe, P, and possibly S in the body fossils, which contrast with the*
*high concentrations of Mg, Al, K, Ti, and Ba in the matrix (Table 1, Supplementary Fig.*
*7). The high concentration of Fe in the body fossil indicates the exceptional preservation*
*of soft-bodied fossils because pyrite, formed by iron reacting with sulfates, can quickly*
*replace the soft tissues at the early stage of decay 4, 21, 40, 41, allowing the*
*preservation of soft tissues. The low proportion of pyrite in fonds makes the Tongshan*
*Biota most comparable to the preservation condition of the middle Cambrian Burgess*

*Shale Lagerstätte (Supplementary Discussion). In addition, the Tongshan Lagerstätte,*
*preserved in siliceous shales/mudstones dominated by silicate minerals, likely benefits*
*from the rapid precipitation of silica cements, which has been an important factor for the*
*preservation of mouldic Ediacara Lagerstätte 20.” in lines 179-191 and “Thin sections of*
*shales/mudstones show parallel laminae composed of thick clay-rich background beds*
*and thin silt-rich event beds (Supplementary Fig. 8). The thick clay-rich beds indicate a*
*prolonged period of low-energy condition, alternating with short duration silt-rich beds*
*formed under relatively higher energy. Periodic fluctuations in local environmental*
*energy resulted in the rapid burial of organisms. In addition, these parallel beds lacking*
*sedimentary features such as cross-stratification indicate that the deposition occurred*
*below the storm wave base.” in lines 195-202.*

The detailed discussion is also organized as one subsection in the Supplementary
information: “**Supplementary discussion – Geochemical and sedimentological**
**evidence for the palaeoenvironment of the Tongshan Lagerstätte**

*EDS analyses of soft-bodied fronds (Table 1, Supplementary Fig. 7A–W) display*
*high concentrations of C, Fe, P, and possibly S, which are contrasting to the high*
*concentrations of Mg, Al, K, Ti, and Ba in the matrix. Si and O have similar*
*concentrations in both body fossils and the matrix. Aluminosilicates could partially fill the*
*voids left by soft tissues in BST fossils 50-54. This explains why the body fossils have*
*lower contents of aluminosilicate minerals than the matrix and indicates that the high*
*concentrations of C, Fe, P, and possibly S, are true signals of body fossils. The presence*
*of Fe in the body fossil is a direct indicator for exceptional preservation of soft-bodied*
*fossils because irons react with sulfates to form pyrites, which can quickly replace the*
*soft tissues at the early stage of decay 53, 55-57, allowing the preservation of soft*
*tissues. During subsequent weathering, original pyrites could form iron oxides and, at*
*the same time, the sulfuric acids released due to this alteration were washed away. This*
*explains the lower content of S detected in the body fossils and the reddish color of the*
*fossils. The presence of S further indicates the microenvironments around body fossils*
*during burial were partially anoxic, allowing iron in the water columns to react with the*
*sulfates surrounding the decay materials. The concentration of pyrites varies among*
*Lagersätten. In the two most well-known BST Lagersätten, specimens of the middle*
*Cambrian Burgess Shale Biota contain a lower proportion of pyrite compared to those of*
*the lower Cambrian Chengjiang Biota 58. The low proportion of pyrite in fossils makes*
*the Tongshan Biota most comparable to the preservation condition of Burgess Shale*
*Biota.*

*In addition, the Tongshan Lagerstätte is preserved in siliceous shales/mudstones*
*dominated by silicate minerals, which distinguishes it from most BST biotas preserved*
*by early carbonate cements 59. This condition is likely related to the widespread*
*weathering of continental crust near the terminal Ediacaran 60-62. While the rapid*
*precipitation of silica cements has been an important factor for the preservation of*
*mouldic Ediacara Biota 63, the Tongshan materials from the siliceous deposits likely*

*benefit from this mechanism. The BST Tongshan Lagerstätte shares preservational*
*mechanisms not only with the typical Burgess Shale Lagerstätte but also with the*
*Ediacara Lagerstätte, representing an intermediate style that bridges the Ediacara-type*
*and BST preservation and contributes to a better understanding the early evolution of*
*life.*

*The dark-coloured, thick clay-rich layers represent slow sedimentation of clay*
*particles from the water column, corresponding to the “background” beds, while the light-*
*coloured, thin silt-rich layers represent the “event” beds (Supplementary Fig. 8A–C). Cyclic*
*changes between clay-rich and silt-rich beds reflect periodic fluctuations in local*
*environmental energy, resulting in the rapid burial of organisms. These parallel beds*
*without cross-stratification indicate that the deposition occurred below the storm wave*
*base. Many well-preserved BST Lagerstätten such as Qingjiang 64 show similar*
*depositional phenomena.”*

Minor comments:

“Unlike many Cambrian Lagerstätten, Burgess Shale-type (BST) preservation is rare and
moldic Ediacara-type preservation provides insight into the early evolution of metazoans
including typical fronds” (line 15); this sentence appears to be saying two different things
– I think it needs re-wording.

Response:

We revised it to become “Because the informative Burgess Shale-type preservation is rare
in the Ediacaran, mouldic Ediacara-type preservation provides insight into the early
evolution of metazoans including typical fronds”. This highlights the rarity of BST
preservation in the Ediacaran, and our work provides such preservation for early animals.

“The large surface area of the new algae might have locally increased oxygen
concentration via photosynthesis to facilitate metazoans” (line 87); this reads a little
strangely, and feels like an unsupported statement. I suggest adding a reference, or
removing if there is no previous work to support this being important in Ediacaran
ecosystems.

Response:

This sentence had been deleted from the main text.

“...whether a stem is present remains unclear but present, the stem must have been short”
(line 144); likewise, this sentence reads a little strangely...do the authors mean to write,

'if a stem was originally present, it must have been short'?

Response:

Right. We have corrected this sentence to become "...whether a stem is present remains
unclear. **However, if a stem was originally present**, it must have been short."

Reviewer #2 (Remarks to the Author):

At present, the submitted paper describes Burgess Shale type preservation as that ‘which
preserves original information on organic integument and internal soft tissues, in addition
to hard exoskeletons.’ The paper then states ‘Unlike Cambrian Lagerstätten with BST
preservation, the Ediacara Biota 43 features Ediacara-type preservation that preserves
the outline of structures, but the original compositional information is unavailable.’

It is correct that many Ediacaran biota fossils are preserved as moulds or casts. However,
while rarer than mouldic preservation, organic preservation of Ediacaran fossils is known
for example from the White Sea assemblage. E.g. see Bobrovskiy et al 2019 Nature
Ecology and Evolution:

<https://www.nature.com/articles/s41559-019-0820-7>

‘Preservation of organic matter plays an important role in the taphonomy of all fossils from
the White Sea localities. A close investigation of dozens of Ediacaran macrofossils,
including Dickinsonia, Andiva, Kimberella, Aspidella, Sabellitides and Beltanelliformis
collected from unweathered layers in the White Sea revealed that they all are preserved
organically, including those that occur within sandstone or clay (Fig. 6a–d)6,7,45. ‘

Response:

We appreciate that you mentioned one of the breakthroughs among the Ediacara-type
fossils in recent years. We fully acknowledge the groundbreaking work of Bobrovskiy et
al. (2019) demonstrating organic preservation in White Sea Ediacaran fossils and many
other related works that have been already cited in line 259 of our paper. However, we
wish to clarify several key distinctions. The White Sea examples represent a specialized
case of organic preservation within Ediacara-type taphonomy (primarily in coarse-grained
sandstones). Our Tongshan material exhibits true BST preservation (carbonaceous
compression in fine-grained mudstones/shales). The Tongshan Lagerstätte provides the
first clear evidence of BST preservation for Ediacaran frondose organisms. This
represents a distinct taphonomic pathway from both classic Ediacara-style preservation
and the Ediacara-style preservation with the organic preservation. The BST Tongshan
Lagerstätte bridges the gaps between traditional Ediacara-type preservation, Ediacara-
type preservation with organic remains (as seen in Bobrovskiy et al. 2019), and BST
preservation. It also helps bridge the taphonomic continuum between Ediacaran and
Cambrian preservation styles. Our findings complement rather than contradict previous
work. We have modified the relevant text in the last paragraph of the Discussion, added
the Supplementary Figure 10 to more precisely articulate these distinctions, and included
references to Bobrovskiy et al.'s important work.

See also Cai et al 2012 Palaeo³:

<https://www.sciencedirect.com/science/article/pii/S003101821200082X>
'Burgess Shale-type (BST) fossils often are preserved as two-dimensional carbonaceous
compressions, sometimes aided by two mineralization processes: pyritization and
aluminosilicification, defined by a thin and sometimes localized coating of authigenic pyrite
or aluminosilicate minerals on the carbonaceous materials. Here we report similar
mineralization modes within the late Ediacaran Gaojiashan Lagerstätte of southern
Shaanxi Province, South China.'

Therefore the manuscript should be updated in the light of known Ediacaran organic
preservation.

**Response:**

We completely agree with the reviewer that BST preservation does occur in the Ediacaran,
as demonstrated by Cai et al. and others. Our study complements these previous findings
by documenting the first known case of BST preservation specifically for frondose
Ediacaran organisms. We have considered the discussion of the Gaojiashan biota (Cai et
al. 2012) into our treatment of Ediacaran BST preservation.

The work of MacGabhann (MacGabhann, 2014, *Geoscience Frontiers*, 5, 53-62)
provided a comprehensive summary about the preservation of Ediacara Biota (fig. 2 in
MacGabhann, 2014). Although other modes of preservation exist among Ediacaran
organisms, the "moulds+casts (fig. 2 in MacGabhann, 2014) is the dominated one in the
Ediacara Biota. The frondose metazoans such as rangeomorphs are uniquely preserved
as the moulds and casts. Our work extends the preservation of these typical Ediacaran
metazoans with BST mode. Following your suggestion, we also have added a new
Supplementary Figure S10 to better illustrate: Panel A: The traditional paradigm of
Ediacara Biota and frondose animals; Panel B: How our findings expand this
understanding.

The important carbonaceous and/or organic preservation related to fronds, the main
focus of this work, is discussed in the main text. Following your suggestions, we have
added new content to the final paragraph of the main text to clarify this issue: "*Laboratory
simulations have indicated that fossils of the Ediacara Biota preserve not only the external
morphology but also the morphology of soft external or internal organic "skeletons" 58.
Also, presence of organic matters has played an important role in the preservation of fine
details of early metazoans through the Ediacara-type preservation 58. Analytical
techniques such as biomarkers and stable isotopic data have helped to solve long-
standing debates by revealing original compositional information of tissues 59-63. The
BST preservation of the Ediacara Biota reinforces these observations and offers greater
opportunities to examine internal "skeletons" or structures in detail, devoting to solve long-
standing debates related to the early evolution of metazoans.*"

However, for other BST preservation of other Ediacaran organisms, we had
concluded in a subsection "**Supplementary discussion – BST preservation in the**

**Ediacaran period** ” in the supplementary information. Cai et al.’s work (2012) is also cited
in the line 298 in this section.

Unfortunately, the majority of the figure photographs do not show sufficient detail of the
specimens to confirm their taxonomic attributions.

**Response:**

We recognized that using a polarized filter for photograph is a problem in showing the
details of fronds in the previous version. All frond specimens have been completely
rephotographed using low-angle lighting to enhance topological features. We have
included a new specimen in Figure 4 that particularly well demonstrates the characteristic
features, and we have carefully re-examined all specimens in Figures 5-6 to ensure their
diagnostic features are now clearly visible. For particularly delicate structures, we have
included detailed close-up insets where appropriate and provided multiple viewing angles
for key specimens. The improved photographs better reveal the fractal branching patterns
characteristic of rangeomorphs and fine structural details of putative holdfast regions.

For example the manuscript states ‘Ten fan- and bush-like specimens belong to Bradgatia
104 sp. (Fig. 4a–b and Extended Data Fig. 7a–f) are one of the youngest appearances of
this genus^{12,28} 105 . Segments of branches are clearly visible (Extended Data Fig. 106
7a–f) and show 3D in morphology (Extended Data Fig. 7e–f).’

However, figure 4 a and b show circular specimens with some irregular lines. If we
compare these with Ediacaran specimens with exceptional preservation, that show their
taxonomically distinctive features it is very difficult to confirm whether they may be of the
same taxon or not.

**Response:**

We have removed all specific genus-level assignments (e.g., Bradgatia) from the
manuscript. The specimens are now conservatively comparable to the rangeomorphs. The
description now focuses on general rangeomorph characteristics rather than specific
taxonomic features. In addition, comparative discussions have been refined to highlight
similarities in overall morphology rather than specific taxonomic identities. We have
enhanced the figures using with multiple angle-lighting to better reveal the observable
morphological features. We fully agree with the reviewer that making the definitive
taxonomic assignments is challenging, and we sincerely appreciate this opportunity to
strengthen our manuscript’s scientific rigor. These revisions maintain the important
paleontological significance of these specimens while adopting a more appropriately
conservative approach to their interpretation.

Similarly in extended data fig 7 there is a large amount of textural noise in the images and
it is not possible to confirm the presence of, for example, rangeomorph type fractal
branching.

Response:

We have rephotographed all frond specimens without using a polarized filter, and the
structural details are more clearly displayed under low-angle lighting. As the contents have
been reorganized, the details displayed in previous extended data figure 7 have been
moved to the main figures. Please check the new figures (Figs. 4-6) for the fractal
branching, which has been shown in the close-up images.

The fossil shown in Fig 4 c to d is very interesting. However, from these images it is difficult
to confirm more than they show an elongate bag-like organism with at least first order
branches, ribs or annulations.

Response:

This specimen (Fig. 6A-B) was rephotographed under low-angle lighting. Some
secondary-order branches are clearly visible (Fig. 6C). We limit our description and
discussion of this specimen to its interpretation as a rangeomorph.

Fig 4e is compared to Swartpuntia. However, this taxon is notable for having a wide aspect
ratio which the figured specimen does not appear to share. What is the basis for the
suggested attribution to this taxon as opposed to another taxon entirely? The discussed
holdfast is not clearly visible in the figures.

Response:

This specimen (Fig. 6E) raises a concern due to its significantly weathered condition
compared to other materials. We provide a general description of it and state that it is a
frond animal in line 157.

Overall, I would suggest that there are two options available to the authors. First, either to
improve, if possible, the images used to support the taxonomic attributions and to further
explain and evaluate the evidence for specific attributions. Second, to make more tentative
taxonomic attributions in the present paper. Perhaps a combination of these options could
be pursued.

Response:

We have carefully implemented both recommended approaches to strengthen our
manuscript. All frond specimens have been rephotographed using optimized techniques:
removed polarizing filters that previously reduced clarity; applied low-angle lighting to
enhance topological features; and captured multiple views of key specimens. The new
images (including the additional specimen in Fig. 4) now more clearly show first- and
second-order branching patterns. We have also adopted more tentative taxonomic
attributions throughout: removed all species-level identifications, focused on broader
rangeomorph affinity rather than precise classification. The results and discussion now
emphasize observed morphological features rather than taxonomic assignments,
comparisons with known rangeomorph characteristics, and the significance of BST
preservation for these Ediacaran forms. We have also provided an additional specimen
(Fig. 4) as particularly clear examples of diagnostic rangeomorph branching patterns,
three-dimensional preservation of frond elements, and consistent morphology across
multiple specimens. We believe these revisions have significantly strengthened the
manuscript by providing better visual documentation of key features, adopting more
appropriate taxonomic caution and maintaining the important scientific implications of
these finds.

Reviewer #3 (Remarks to the Author):

In the Abstract (line 28) the authors state that their fossils will provide 'A new repository of
original compositional information'. Likewise in the Introduction (line 49) 'This study ...
makes compositional information widely available to the Ediacaran metazoans such as
fronds' and Discussion (final line 238) 'the preservation ... will reveal more information on
the original composition of Ediacaran metazoans'. However, this claim, which the authors
appear to consider central to the significance of their paper, is nowhere supported by any
chemical data. Note also that the composition will not be 'original' but diagenetically altered
to longer chain compounds by polymerization.

Response:

We appreciate your insightful comment regarding our use of the term “original
compositional information.” We acknowledge that diagenetic alteration (e.g.,
polymerization of organic compounds) modifies the primary chemistry of fossils. However,
Burgess Shale-type (BST) preservation uniquely retains both morphological and
geochemical evidence that reflects original biological structures, even if the organic
molecules themselves have undergone diagenetic transformation.

For illustrating this issue and background information better, we revised the
description and expanded the explanation of original compositional information for BST
preservation in the introduction as “..., which preserves compositional information in both
chemical and morphological aspects of organic integuments such as cuticle details 6, 7
and internal soft tissues such as muscles, gut systems, and nervous systems 8-11, in
addition to hard exoskeletons” in the introduction. The word “Original” is removed from this
sentence. In the revision, we emphasize that the BST preservation retains the
compositional information in both chemical and morphological aspects. Chemical
information represents only one part of the original information and includes the modified
compounds you mentioned. The morphological information includes both external and
internal structural details, such as the morphology of gut and gut glands, nervous systems,
brains, different layers of cuticle, and internal details of spines.

We here use the spine as an example to explain the difference between the Ediacara-
type and BST preservation. In Ediacara-type preservation, the spine is left only with an
empty envelope-like morphology, such as external or internal moulds (e.g., *Coronacollina*
*acula* in Clites et al. 2012, *Geology*, 40, 307-310). In contrast, in BST preservation, the
spine or claw can show multiple internal structural layers (e.g., *Hallucigenia sparsa* in
Smith and Ortega-Hernandez, 2014, *Nature*, 514, 363-366), which can further be
distinguished based on their chemical compositions, appearing as dark- and light-colored
regions. This represents a key advantage of BST preservation. Discovering more
Ediacaran metazoans with BST preservation would provide access to additional structural
details, offering great insights into the early evolution of metazoans.

To present the chemical composition, we have added Table 1 and Supplementary

Figures S7 and S8 to display the preservational information, which is organized as a
subsection in the Supplementary information: “*Supplementary discussion -*
*Supplementary discussion – Geochemical and sedimentological evidence for the*
*palaeoenvironment of the Tongshan Lagerstätte*”. Main text also includes this information
in lines 179-202.

In the revised Abstract, we have modified the last sentence to become “The rarity of
Burgess Shale-type preservation of fronds reflects the rarity of fine-grained deposits in the
Ediacaran Period and expands the Ediacara biota with an informative repository of tissue
and integument composition”.

In summary, while we agree that “original composition” must be interpreted through
the lens of diagenesis, BST fossils nonetheless provide the closest available record of
Ediacaran tissue systems. Our revisions clarified terminology, incorporate new
geochemical data, and reframe the significance of BST preservation for early metazoan
evolution.

'Bat-shaped', 'tie-shaped' (lines 83,84), and 'corn-like' (line 136) are not useful descriptors
as they will suggest different comparisons to different people. Likewise 'meandering-like'
(line 94) is not going to prompt images of this specimen.

Response:

For better illustrating these shapes, we have reworded these descriptions with more
details.

- 1) A tie-shaped form (3F) is corrected as “an irregular circular form”.
- 2) “A bat-shaped form” (3E) is corrected to become “a baseball bat-like form with an
expanded distal end”.
- 3) “A corn-like outline” (5G) is modified as “...a spherocylindrical outline...”.
- 4) We have deleted the word “meandering-like” and replaced the description of the
specimen (Fig 3B) with “A tubular structure showing segment-like divisions, ...”.

The surface area (line 88) would only be a significant source of oxygen if the algae
occurred in dense groups. Is there any evidence of this?

Response:

We have deleted this sentence from the main text after carefully reviewing the weakness
of its argument.

Does 'undefined' (line 94) simply mean that the specimen in 3b is poorly preserved.
Likewise (line 99) how confident are you that this specimen is an open tube or that the

irregular opening represents the true termination.

Response:

Regarding the specimen in Fig. 3B, the term "undefined" refers to certain morphological
features being less distinct due to taphonomic factors, not necessarily poor overall
preservation. The specimen retains diagnostically important characteristics despite partial
obscuration. The key features supporting our identification include segment-like divisions
(marked with brackets), elliptical segment outlines characteristic of cloudiniids, and cone-
in-cone construction at the distal end (matrix-filled areas). These features are visible in
new photograph taken without polarization to minimize optical artifacts and allow us to
make the argument that it is comparable to cloudiniids.

We agree the apparent opening at top may not represent the true termination. We
have modified the description to: "A tubular structure showing segment-like divisions and
its distal portion showing irregular opening at top may be related to cnidarians or
annelids..."

The designation 'sp.' means that the material preserves features diagnostic of the genus
yet cannot be assigned to a known species. If you use 'sp.' you need to justify its
assignment to the genus, but also explain how it differs from other species of the genus in
question. You achieve this to some extent in the case of *Swartpuntia* (line 126 ...) but not
for *Bradgatia* (line 103 ...).

Response:

We have toned down our results and discussions. Descriptions of frond materials are
restricted to general features that are characteristic of the Rangeomorpha. All discussions
of lifestyles and occurrences for *Bradgatia* and *Swartpuntia* have been removed. Thus,
we maintain our focus on the BST fronds rather than assigning them to a particular genus
or species. The revised manuscript presents these specimens as important examples of
BST-preserved rangeomorphs without assigning them to a particular genus or species.

Fresh algae and animals can endure significant transport without damage. The
preservation of holdfasts does not necessarily demonstrate that the organisms are buried
in situ. Is there sedimentological evidence to support 'quick burial and in situ preservation'
(line 179)?

Response:

We integrate multiple lines of evidence to substantiate the "quick burial and in situ
preservation".

1) *Beltanelliformis* colonies (Fig. 2H) exhibit closely spaced but non-overlapping

individuals, consistent with undisturbed growth rather than postmortem accumulation. This
spatial arrangement is a hallmark of in situ preservation in Ediacaran settings. Another
example is that the holdfast of one new frond (Fig. 6D) is preserved parallel to bedding,
with morphological features (e.g., a disc standing out from the central stalk or stem)
suggesting it is originally anchored to the seafloor, rather than transported.

2) A high concentration of pyrite in the frond body (Table 1; Supplementary Figure 7)
indicates localized microbial sulfate reduction in a confined microenvironment—a process
requiring rapid organic matter burial to prevent oxidative degradation.

3) Laminated clay-silt alternations in thin sections (Supplementary Figure 8) reveal that
background sedimentation (clay-rich laminae) reflects prolonged low-energy conditions,
allowing undisturbed microbial mat development and organism colonization; event beds
(silt-rich laminae) represent episodic, slightly higher-energy pulses that rapidly
entombed benthic communities without erosive disruption; absence of cross-stratification
or scour features suggests minimal reworking, supporting the interpretation of sudden
burial.

In conclusion, the combined biological, geochemical, and sedimentological data align
with Burgess Shale-type (BST) taphonomic models, where event-driven rapid burial
facilitated exceptional preservation of in situ communities.

We organized the geochemical and sedimentological evidence as one subsection
(lines 207-249), “**Supplementary discussion – Geochemical and sedimentological**
**evidence for the palaeoenvironment of the Tongshan Lagerstätte**”, in the
Supplementary information.

In the main text, we added “*EDS analyses of two frond specimens display high*
*concentrations of C, Fe, P, and possibly S in the body fossils, which contrast with the high*
*concentrations of Mg, Al, K, Ti, and Ba in the matrix (Table 1, Supplementary Fig. 7). The*
*high concentration of Fe in the body fossil indicates the exceptional preservation of soft-*
*bodied fossils because pyrite, formed by iron reacting with sulfates, can quickly replace*
*the soft tissues at the early stage of decay 4, 21, 40, 41, allowing the preservation of soft*
*tissues. The low proportion of pyrite in fonds makes the Tongshan Biota most comparable*
*to the preservation condition of the middle Cambrian Burgess Shale Lagerstätte*
*(Supplementary Discussion). In addition, the Tongshan Lagerstätte, preserved in siliceous*
*shales/mudstones dominated by silicate minerals, likely benefits from the rapid*
*precipitation of silica cements, which has been an important factor for the preservation of*
*mouldic Ediacara Lagerstätte 20.” in lines 179-191 and “Thin sections of*
*shales/mudstones show parallel laminae composed of thick clay-rich background beds*
*and thin silt-rich event beds (Supplementary Fig. 8). The thick clay-rich beds indicate a*
*prolonged period of low-energy condition, alternating with short duration silt-rich beds*
*formed under relatively higher energy. Periodic fluctuations in local environmental energy*
*resulted in the rapid burial of organisms. In addition, these parallel beds lacking*
*sedimentary features such as cross-stratification indicate that the deposition occurred*

*below the storm wave base.” in lines 195-202.*

You need citation(s) to the conventional view that there is no bias (line 202).

*Response:*

*We cite a landmark work (Buatois et al. 2014, Nature Communications, 5, 3544) here to*
*represent the conventional view.*

If you cite 53 on line 237 (Love et al. 2009) you should also cite the alternative more recent
view: Bobrovskiy, I., Hope, J.M., Nettersheim, B.J., Volkman, J.K., Hallmann, C. and
Brocks, J.J., 2021. Algal origin of sponge sterane biomarkers negates the oldest evidence
for animals in the rock record. Nature Ecology & Evolution, 5(2), pp.165-168.

*Response:*

*By following your suggestion, we cited Bobroskiy et al. (2021) in this sentence.*

Manuscript ID: **NCOMMS-25-02355-A**

We thank the reviewers once again for their time and valuable feedback. The
current revision, including an addition of supplementary data and a reorganization
of text, has been made in response to these comments.

Major improvements:

- 1) Additional EDS analysis has been conducted, and the corresponding data
are now presented in the new Supplementary Figure S7, which clearly
shows the separation between the fossil body and the matrix. (Reviewers 1
and 3)
- 2) Abstract has been revised to incorporate more general information on
preservation. (Reviewer 2)
- 3) The caption of Supplementary Figure S10 has been added with new
information to clarify the difference between figure S10A and S10B.
(Reviewer 3)
- 4) Description of EDS analysis has been moved to the Results section.
(Reviewer 3)

Modified text is highlighted with a yellow background and point-to-point responses
to comments are showing below.

23 24 **REVIEWER COMMENTS**

Reviewer #1 (Remarks to the Author):

The paper is substantially improved - the authors have backed off the specific
genus designations, and have presented some new specimens which are more
convincingly rangeomorphs. I am also grateful that elemental maps are shown (in
Supp. Fig. 7), which give valuable context. If I was being nitpicky, the figure doesn't
seem to present maps of fossils vs. matrix; some important information is thus
hidden, and preventing a proper comparison. Ideally, one of the maps would
encompass the border between the fossil and matrix, so enrichments in specific
elements are clearer.

**Response:**

Thank you for highlighting this point, which helps to make the chemical data more
clearly visible. We have conducted an additional analysis that includes both the
fossil body and the matrix in each elemental map. Please refer to the new
Supplementary Figure S7.

Otherwise, I think this is close to being publishable. Two final comments are below:

"pyrite, formed by iron reacting with sulfates, can quickly replace the soft tissues at the early stage of decay, allowing the preservation of soft tissues" (line 183); the authors may wish to cite Gibson et al. (2023) in *Geobiology*, who were able to experimentally replicate the formation of pyrite in association with decaying diploblasts and triploblasts.

Response:

We appreciate you bringing up this experimental work, which is closely related to our study. We have cited this work at the end of this sentence on line 191.

"BST preservation of typical fronds indicates that Ediacara-type preservation, which largely occurs in coarse grained deposits such as sandstones, is restricted by available rocks" (line 239); similar to the comment above; Gibson et al. (2023) showed that the iron content of sediments is likely also an important control on Ediacaran-style preservation, and which hints at secular and paleogeographic biases.

Response:

We appreciate this evidence in helping to decipher the preservation condition of Ediacaran organisms. We have added following sentence to lines 254-257: "As the iron weight percentage of sediment (iron:sediment) plays an important role in Ediacara-type preservation and is generally higher in shales than in sandstones⁴², preservation bias for Ediacaran organisms is further highlighted by the rarity of available deposits, such as mudstones from the Ediacaran period."

Reviewer #2 (Remarks to the Author):

As detailed in their point by point response to review the authors have provided
additional information and revised the text of their manuscript describing a
fascinating new fossil biota. However, I would still suggest they go further in re-
framing their discoveries to be less dependent on specific affinities of some of the
fossils to Ediacaran taxa, which remain uncertain. The authors have provided a
new figure 4, which they link to Rangeomorpha. However, given the difficulty in
assessing branching structure from photographs, I am unable to verify the
presence of rangeomorph elements.

Response:

We understand your concerns, as these are the first BST frond materials presented
to us. Notably, considering that the first organism of the Ediacara Biota was
described in 1872, it has been over 150 years during which the Ediacaran fronds
have only been observed with Ediacara-type (or mouldic) preservation.

We followed a systematic procedure to assign these materials to rangeomorphs,
considering the first-round comments from you and the other two reviewers.

Firstly, there are no other contemporary or Ediacaran organisms, except the typical
fronds, that can be compared to these materials. We appreciate your agreement
on this point. Secondly, the fronds can be divided into three major categories:
rangeomorphs, arboreomorphs, and erniettomorphs. **Erniettomorphs**, which
process only one-order branches, can be excluded, as the Tongshan materials
display more than one order of branches. The overall morphology of
Erniettomorphs is also different from most Tongshan materials, with the exception of
the specimen in Figure 6E. **Arboreomorphs** bear “pods” and units (figure 6 in
Dunn et al., 2019, Palaeontology, v. 62, part 5, p. 851-865), and our materials don't
show such traits. The presence of first-order branches and its associated
perpendicular branches makes most Tongshan materials comparable to
**rangeomorphs**. Finally, there is no evidence for us to treat the Tongshan materials
different from these three major categories. In summary, assigning these BST
materials to rangeomorphs is most appreciate classification based on the current
understanding of Ediacaran organisms.

I would suggest rephrasing the text throughout to suggest plausible affinities, and
not to frame the paper in terms of novelty dependent on the more uncertain of
these affinities, which I don't believe is necessary for this to be a very interesting
and publishable paper.

Response:

We really appreciate that you provide us the thoughtful comments, which are
devoted to strengthen this work. If the Tongshan Biota contains only type of fronds,
it would be difficult to make the argument we have presented. However, four types
of fronds have been identified, all displaying similar patterns of branches, the first-
and second-order branches (Figs. 4-6). Multiple lines of evidence including total
morphology support the assignment to rangeomorphs. The branching styles or
characters of these BST fronds don't support an argument that that they are
different from rangeomorphs. We do appreciate this comment, but we believe it is
more appropriate to remain the assignment to rangeomorphs rather than attribute
them to unknown or uncertain organisms.

For example, I would suggest modifying line 17 of the abstract to briefly describe
the whole diversity of fossils they have found and the range of their plausible
affinities e.g. all the protists, algae, metazoans they have identified, and
mentioning apparent fronds within this without focusing on them so heavily.

Response:

We have revised the first sentence to become “.....provides insight into the early
evolution of organisms like metazoans (including typical fronds), protists, and
algae.” The Tongshan Lagerstätte is unique for its BST-preserved fronds, which
represent one of the central arguments in our paper. The final sentence of the
abstract, which has been revised based on your comment below, clearly reflects
the broader significance of the Tongshan Lagerstätte and its BST preservation.
Therefore, we here keep the “typical fronds” in this sentence, as it highlights the
key discovery in this new Lagerstätte.

Instead of 'representing the Rangeomorpha' in the abstract, I would suggest briefly
including some of the more nuanced discussion of the BST preservation which the
authors included in their point by point response e.g. summarising their response:

'Our Tongshan material exhibits true BST preservation (carbonaceous 244
compression in fine-grained mudstones/shales).' 'The BST Tongshan 248
Lagerstätte bridges the gaps between traditional Ediacara-type preservation,
Ediacara249 type preservation with organic remains (as seen in Bobrovskiy et al.
2019), and BST 250 preservation.'

I would suggest similar re-framing throughout.

**Response:**

Thank you for highlighting this detail, which represents the broader implication of
our study and aligns with its central focus. We have revised the final sentence of
the abstract to become “*The rarity of Burgess Shale-typeand bridges the gap*
*between traditional Ediacara-type preservation, Ediacara-type preservation with*
*organic remains, and Burgess Shale-type preservation*”.

Reviewer #3 (Remarks to the Author):

The authors have dealt with two major issues raised by the reviewers. The quality
of the illustrations is greatly improved and a new specimen added. The evidence
for fractals is difficult to demonstrate but is now much clearer. The taxonomic
assignments are more generalized to reflect the nature of the material.

The authors have added EDS data (Figure S7 and Table 1). Table 1 provides
weight per cent data but there is no information in Methods indicating whether
these are based on the area mapped or spot samples.

Response:

We appreciate you pointed out this detail. We have added this information at the
end of Methods section: “All sites were defined with a uniform area of
approximately 1.12 mm x 0.84 mm, the data of which are presented in Table 1. An
additional site, the fourth site, outlined with a red box in each specimen
(Supplementary Fig. 7A, L), displays the difference in elemental distribution
between the frond body and the matrix.”. In the caption of Table 1, we also added
the following sentence “All sites, including site 1, site 2, and the matrix, were
retrieved from a uniform area”.

Unfortunately Figure S7 includes maps of one site on each specimen only whereas
three were mapped. It would be useful to include the maps for the other areas
(particularly the matrix). Needless to say maps convey different information to
weight percent data, specifically element distribution, and overlap of different
elements provides an indication of the minerals present.

Response:

Thank you for pointing out this issue. We have conducted an additional EDS
analysis that includes both the fossil body and the matrix in each elemental map,
which are now showing in the new Supplementary Figure S7. The information
related to Table 1 has also been updated in the method section to reflect the
modifications made to Supplementary Figure S7.

To preserve the specimen in its original state for further analysis, we avoided
coating the specimen. However, this resulted in severe charging issue during SEM
imaging. Thus, the overlapping elemental maps filed to clearly display the element
distribution. For this reason, only individual elemental maps are provided.

Given that the EDS data are important to the interpretation of the preservation
(BST-type or not) that information should be moved from Discussion to Results.
Note that the high concentration of Fe in the body fossil does not, per se, indicate
exceptional preservation, and the evidence for pyrite is weak at best: Fe and S
coincide somewhat in Figure S7 C and K, but not in N and V (indeed S is essentially
absent in V). Likewise the fact that the lithology is dominated by silicates does not
allow a silica (i.e. quartz) cement to be inferred: the silicates are presumably clays
as petrography of the thin sections would confirm. A high concentration of P is
mentioned (line 180) but there is no discussion of its significance and a map of P
is omitted for the first frond specimen in Figure S7!

**Response:**

By following your suggestion, we have moved a part of the EDS content to the end
of the results section: *“EDS analyses of two frond specimens, ESEN 0017 and
ESEN 0019a, display high concentrations of C and Fe in the body fossils, which
contrast with the high concentrations of Mg, Al, K, Ti, and Ba in the matrix (Table
1, Supplementary Fig. 7). In both specimens, Si and O exhibit similar
concentrations in both the fossil body and the matrix. Specimen ESEN 0017
(Supplementary Fig. 7A) has a higher concentration of S in the fossil body,
whereas specimen ESEN 0019a (Supplementary Fig. 7L) has a higher
concentration of P (Table 1, Supplementary Fig. 7)”*.

In the new analysis, Fe and S in specimen ESEN 0017 (Fig. S7A) are matching
with each other, but this relationship is still unclear in specimen ESEN 0019a (Fig.
S7L). In the Table 1, ESEN 0019a (Fig. S7L) has a high concentration of S in the
matrix, whereas ESEN 0017 (S7A) shows a high concentration of S in the frond
body, consistent with the results in figures S7C, K, N, and V. In summary, the new
maps displayed in Figure S7 are consistent with the data summarized in Table 1.
An explanation about the lower concentration of S observed in the analysis is
provided in the Supplementary discussion: *“During subsequent weathering,
original pyrites could form iron oxides and, at the same time, the sulfuric acids
released due to this alteration were washed away. This explains the lower content
of S detected in the body fossils and the reddish color of the fossils.”*,

We have deleted the content about the silicate cementation from both the main
text and the supplementary discussion. Confirming this mechanism requires
further study, as it demands substantially more evidence, including the point you
have mentioned.

“P” was not detected in the first frond specimen, ESEN 0017 (Supplementary Fig.

S7A; Table 1). That is why there is no “P” elemental map for this specimen. We
also expanded the discussion to include phosphate content. New text--“The high
concentration of P in the specimen ESEN 00191a (Table 1, Supplementary Fig. 7L)
provides additional chemical evidence for the early diagenetic mineralization of
soft tissues⁴³. Phosphatization preserves tissues with extremely high fidelity but is
a strongly biased taphonomic process affected by factors such as size and
environment^{44, 45}. Decay of the small-sized frond may produce a suitable
microenvironment for the process of phosphatization.”--has been added to lines
194-200.

Given the thickness of the silt laminae (Figure S8) it is hard to imagine rapid burial
of in situ fronds.

Response:

In well-known Lagerstätten, thickness of silt laminae (or event beds) can also be
very thin, with some reaching millimeter-scale thickness (i.e., fig. 1e, f in Caron et
al. 2014, Nature Communications, 3210; supplementary fig. 1B in Fu et al. 2019,
Science, 1338-1342; fig. 5A in Zhao et al. 2009, Palaios, 826-839). All these
conditions are indicative of rapid burial. In addition, a few more observations are
available from the thin-sections of Tongshan rock samples. 1) The thickness of silt
laminae is different from bed to bed, indicating different durations of silt deposition.
2) Clay-beds and silt-beds alternate with each other, indicating the presence of
cyclic events. 3) The boundaries between clay-beds and silt-beds are very sharp,
indicating that silt beds entombed the lower living community of organisms,
creating a clear separation from the overlying living community. These data support
rapid burial of the Tongshan materials.

Since there are no distinct transportation marks near the silt-rich beds, combined
with other organismal evidence, we had concluded that the burial is in situ. Here,
we accommodated your concern by adopting the phrasing “....rapid, likely in situ
burial” in the text.

The message in Figure S10 is not clear (to this reviewer at least). BST preservation
is already included in the Ediacaran in MacGabhann's original (2014) diagram,
which is identical to A here, so what does this figure add.? And coloring most the
Ediacaran in green in B obscures information rather than adding to it.

Response:

We have modified the caption of Figure S10, with new text marked with a yellow

background. In the revised caption, we define the Figure A as “the traditional view
of Ediacara Biota” and the Figure B as “the new view of Ediacara Biota”.

Figure S10A, the original work in MacGabhann (2014), is used to provide a clear
contrast with the new Figure S10B, which shows that both mouldic and BST
preservations are the general preservation style of the Ediacara Biota. In figure
S10B, this change causes the original yellow color in Figure A to be replaced with
dark green (or chartreuse), represented by number “6” (Ediacaran fossils with both
mouldic and BST preservations). Although the yellow area (i.e. **mouldic**
preservation) in Figure B is reduced, the original information (i.e. both **mouldic**
**and BST** preservations represented by number 6) in the Figure A is actually
expanded. Expanding the Ediacara-type preservation with BST preservation in the
Ediacara Biota represents a central argument of our study.

Finally, even though the authors have removed mention of 'original' chemical
composition Reviewer 3's concerns about the inclusion of phrases such as
'informative repository of tissue and integument composition' and 'compositional
information in both chemical and morphological aspects' remain. This issue needs
to be clarified so that it is obvious that what is meant is the composition of the
fossils, not the composition of the original tissues. This confusion is exemplified in
the authors' response where they state that 'In contrast, in BST preservation, the
spine or claw can show multiple internal structural layers (e.g., *Hallucigenia sparsa*
in Smith and Ortega-Hernandez, 2014, *Nature*, 514, 363-366), which can further
be distinguished based on their chemical compositions, appearing as dark- and
light-colored regions.' That paper includes no chemical data and it is very unlikely
that the dark and light colored regions differ significantly in either their original
(presumably chitin-protein) or diagenetically altered state.

**Response:**

We agree with you that it should be the composition of the fossils. Following your
comments regarding the “composition” issue and incorporating with Reviwer2’s
feedback, we have modified the last sentence of abstract to: “*The rarity of Burgess*
*Shale-typeand bridges the gap between traditional Ediacara-type*
*preservation, Ediacara-type preservation with organic remains, and Burgess*
*Shale-type preservation*”. We believe this change better highlights the broader
significance of this work.

Reviewer #2 (Remarks to the Author):

My only further suggestion is to consider whether there are similarities to Cambrian fronds and to provide a reference to previous work that has identified fronds in BST preservation modes. The classic reference, for example is Conway Morris, S (1993) Ediacaran-like fossils in Cambrian Burgess Shale-type faunas of North America. *Palaeontology* 36, 593–635. This was also recently reviewed in Hoyal Cuthill, J.F., 2022. Ediacaran survivors in the Cambrian: suspicions, denials and a smoking gun. *Geological Magazine*, 159(7), pp.1210-1219.

Response:

We appreciate your mention of both papers. Some BST-preserved fronds occur in the Cambrian, but they are widely regarded as Ediacaran-like fronds or problematica. For this reason, using the preservation of debated Cambrian organisms to discuss typical Ediacaran fronds is neither adequate nor convincing. This is also one of the reasons the discovery of the Tongshan Lagerstätte is significant. To provide a clear clarification, we added the following text in the section (Expansion of the Ediacaran Biota with Burgess Shale-type preservation) : Some BST fronds preserved in the Cambrian, which lack typical characters such as first- and second-order branches, are cautiously described as Ediacaran-like fronds⁵⁷ or problematica⁵⁸. The preservation of these debated Cambrian organisms^{58, 59} fails to provide solid evidence for the preservation of those typical Ediacara fronds.”

The detailed comparison between the Ediacaran-like fronds in the Cambrian and Tongshan materials requires further study, which is beyond the scope of our current work. However, following your suggestion, we made a further conclusion regarding this discovery. We added “as well as between the Ediacaran-like fronds in the Cambrian and typical Ediacaran fronds.” in the sentence “The material described here extends documentation of BST-style preservation into other elements of the Ediacara Biota, demonstrating a continuation from the Cambrian organisms to the Lantian Biota^{14, 64} as well as between the Ediacaran-like fronds in the Cambrian and typical Ediacaran fronds.”.

In this revision, we cited three more references.

57. Conway Morris S. Ediacaran-like fossils in Cambrian Burgess Shale-type faunas of North America. *Palaeontology* 36, 593–635 (1993).

58. Hu S, Zhao F, Liu AG, Zhu M. A new Cambrian frondose organism: ‘Ediacaran survivor’ or convergent evolution? *J Geol Soc* 180, jgs2022–2088 (2023).

59. Hoyal Cuthill JF. Ediacaran survivors in the Cambrian: suspicions, denials and
a smoking gun. *Geol Mag* **159**, 1210–1219 (2022).
